# Unravelling the physical and physiological basis for the solar-induced chlorophyll fluorescence and photosynthesis relationship using continuous leaf and canopy measurements of a corn crop

Peiqi Yang[1], Christiaan van der Tol[1], Petya K. E. Campbell[23], Elizabeth M. Middleton[4]

[1]Faculty of Geo-Information Science and Earth Observation (ITC), University of Twente, Enschede, 7500 AE, The Netherlands
[2]Joint Center for Earth Systems Technology (JCET), University of Maryland Baltimore County, Baltimore, MD 21228, USA
[3]Biospheric Sciences Laboratory, NASA Goddard Space and Flight Center, Greenbelt, MD 20771, USA.
[4]Emeritus of Biospheric Sciences Laboratory, NASA Goddard Space and Flight Center, Greenbelt, MD 20771, USA.

*Correspondence to*: Peiqi Yang (p.yang@utwente.nl)

**Abstract.** Estimates of the gross terrestrial carbon uptake exhibit large uncertainties. Sun-induced chlorophyll fluorescence (SIF) has an apparent near-linear relationship with gross primary production (GPP). This relationship will potentially facilitate the monitoring of photosynthesis from space. However, the exact mechanistic connection between SIF and GPP is still not clear. To explore the physical and physiological basis for their relationship, we used a unique dataset comprising continuous field measurements of leaf and canopy fluorescence and photosynthesis of corn over a growing season. We found that, at canopy scale, the positive relationship between SIF and GPP was dominated by absorbed photosynthetically active radiation (APAR), which was equally affected by variations in incoming radiation and changes in canopy structure. After statistically controlling these underlying physical effects, the remaining correlation between far-red SIF and GPP due solely to the functional link between fluorescence and photosynthesis at the photochemical level was much weaker ($\rho = 0.30$). Active leaf-level fluorescence measurements revealed a moderate positive correlation between the efficiencies of fluorescence emission and photochemistry for sunlit leaves in well-illuminated conditions but a weak negative correlation in the low-light condition, and which was negligible for shaded leaves. Differentiating sunlit and shaded leaves in the light use efficiency (LUE) models for SIF and GPP facilitates a better understanding of the SIF-GPP relationship at different environmental and canopy conditions. Leaf-level fluorescence measurements also demonstrated that the sustained thermal dissipation efficiency dominated the seasonal energy partitioning while the reversible heat dissipation dominated the diurnal leaf energy partitioning. These diurnal and seasonal variations in heat dissipation underlie, and are thus responsible for, the observed remote sensing-based link between far-red SIF and GPP.

## 1 Introduction

For our understanding of the Earth's climate, estimates of the gross carbon uptake by terrestrial ecosystems are crucial (Falkowski et al., 2000; Friedlingstein, 2015; Solomon et al., 2009). Despite considerable progress in measurement systems and models, contemporary estimates of the gross terrestrial carbon uptake still exhibit large uncertainties (Ryu et al., 2019). On the one hand, eddy covariance flux towers provide point measurements of carbon uptake at selected locations on all continents, which can be used to estimate gross primary production (GPP), but such *in situ* measurements are sparse. On the other hand, optical remote sensing provides spatially continuous and dense data, but these observations are only indirectly related to GPP. In this respect, the development of sun-induced chlorophyll fluorescence (SIF) measurement techniques from satellites has raised expectations. This is because chlorophyll fluorescence (ChlF) as a by-product of photosynthesis has long been used as a probe of photochemistry in laboratory and field studies (Mohammed et al., 2019). Ever since satellite SIF data products related to the far-red fluorescence peak became available during the past decade, numerous studies have reported a strong correlation between far-red SIF and GPP at the local, regional and global scales (e.g., Campbell et al., 2019; Damm et al., 2015; Guanter et al., 2014; He et al., 2017; Wieneke et al., 2016). This SIF-GPP link has been employed to estimate photosynthetic capacity (e.g., Zhang et al., 2014) and crop yield (e.g., Guan et al., 2016).

The rising expectations of far-red SIF rely on a contestable closer relationship with GPP than other optical remote sensing signals, such as well-chosen reflectance indices (Damm et al., 2015; Mohammed et al., 2019; Wieneke et al., 2016). In order to make use of SIF quantitatively, it is necessary to understand the physical and physiological meaning of SIF, and to establish mechanistic understanding of its relation to GPP (Gu et al., 2019; Magney et al., 2019; Miao et al., 2018; Yang et al., 2015). In recent studies, the light use efficiency (LUE) model of Monteith (1977) has been the common starting point for describing GPP and SIF as a function of the absorbed photosynthetically active solar radiation (APAR):

$$\text{GPP} = \text{iPAR} \cdot \text{fAPAR} \cdot \Phi_{Pcanopy} \qquad (1a),$$

$$\text{SIF} = \text{iPAR} \cdot \text{fAPAR} \cdot \Phi_{Fcanopy} \cdot f_{esc} \qquad (1b),$$

where iPAR denotes the available incoming photosynthetically active radiation for a vegetation canopy; fAPAR is the fraction of APAR absorbed by green vegetation; and $\Phi_{Pcanopy}$ and $\Phi_{Fcanopy}$ describe the canopy-scale light use efficiencies for photochemistry and fluorescence, respectively, which are related to the plant physiological status. $f_{esc}$ is the fraction of the emitted far-red fluorescence that escapes the canopy in the viewing direction (per solid angle), which depends on the viewing and illumination geometries and canopy structure (Porcar-Castell et al., 2014; Yang et al., 2020; Yang and Van der Tol, 2018).

From the LUE model, it is evident that the common terms iPAR and fAPAR are primarily responsible for the often-reported linear relationship between SIF and GPP (Campbell et al., 2019; Dechant et al., 2020; Miao et al., 2018; Rossini et al., 2010;

Yang et al., 2018). The combined contribution of $\Phi_{Fcanopy}$ and $f_{esc}$ to the SIF-GPP relationship is much less clear. It has been argued that $\Phi_{Fcanopy}$ may also contribute to the positive correlation between GPP and far-red SIF, while $f_{esc}$ is viewed as an interfering factor. Guanter et al. (2014) implicitly assumed that a positive relationship between $\Phi_{Fcanopy}$ and $\Phi_{Pcanopy}$ exists

and that $f_{esc}$ in the near-infrared region is isotropic and close to unity when explaining the SIF-GPP relationship at the satellite level. However, these assumptions need to be verified, and we still lack a clear conclusion on the physical and physiological basis for the relationship between far-red SIF and GPP.

Dechant et al. (2020) explored the relationship between SIF and GPP for three in situ crop datasets. They found that correcting

SIF for canopy scattering ($f_{esc}$) improved the correlation between SIF and APAR but not GPP. Furthermore, they reported that their estimates of physiological SIF yield ($\Phi_{Fcanopy}$ = SIF/APAR/$f_{esc}$) showed no clear seasonal patterns and were unlikely to contribute to the positive correlation between GPP and far-red SIF. In contrast, Qiu et al. (2019) reported that the similar correction of SIF for canopy scattering resulted in a better correlation to GPP, and Yang et al. (2020) showed that the estimates of canopy-scale light use efficiency of fluorescence ($\Phi_{Fcanopy}$) were clearly higher in young and mature stages than for the

senescent stages, and were correlated with $\Phi_{Pcanopy}$. The inconsistent findings could partly be caused by considerable uncertainties in the estimates of $f_{esc}$ and $\Phi_{Fcanopy}$, especially since the physiological indicators ($\Phi_{Fcanopy}$ and $\Phi_{Pcanopy}$) are still contaminated by canopy structural effects (Yang et al., 2020).

More fundamental understanding can be obtained by returning to the established physiological methods of *in vivo* active

fluorescence measurements to discern the relative energy distribution among the four pathways in plants via photosynthesis, fluorescence and heat losses (both sustained and reversible). At the photochemical level in leaves, it is clear that a change in fluorescence emission efficiency can be attributed to a change in the combined efficiencies of photochemistry and heat dissipation, expressed as photochemical quenching (PQ) and non-photochemical quenching (NPQ) of fluorescence (Baker, 2008; Maxwell and Johnson, 2000). The relationship between the photochemical-level photosynthetic light use efficiency ($\Phi_P$)

and fluorescence reduction (i.e., quenching) was described with the Genty equation as ($F_m-F_s$)/$F_m$ (Genty et al., 1989) which compares the relative fluorescence change from a steady state ($F_s$) to its maximal level ($F_m$) when the photochemical pathway is completely inhibited (e.g., by using a saturating light) . Semi-empirical generalized relationships have further been developed to model these maximal and steady-state fluorescence levels as a function of photosynthetic light use efficiency and temperature (Rosema et al., 1991; Van Der Tol et al., 2014). However, the universal applicability of the latter models has not

been validated, and continuously collected field measurements of active fluorescence at the leaf level along with canopy photosynthesis and SIF measurements are rare, which limits our understanding of their relationship in natural conditions.

The present study aims to assess the drivers of the apparent SIF-GPP relationship using independent measurements of all terms in the light use efficiency model (Eq. 1), collected under different illumination conditions and at different growth stages, at the

leaf and canopy levels. We chose a corn crop (*Zea mays* L.), also referred to as maize, because it provides a relatively simple canopy, typically a row crop with plants nominally having a spherical shape. As a C4 species, corn does not lose carbon through photorespiration, which makes GPP observations from flux towers more representative to the actual photosynthesis of the canopy. Maize is also a globally important crop that comprises the "bread-basket" to feed the world. Some have claimed (e.g., Guanter et al., 2014) that the observed far-red SIF obtained from space reveals that the US cornbelt achieves the highest carbon sink of any of Earth's ecosystems. On that basis alone, and because of the importance of agricultural surveys from space for food security reasons, we are justified to conduct a more comprehensive examination of the photosynthetic function and associated fluorescence activity of this crop, and encourage more such studies of important crops affecting food security.

We drew upon a unique dataset comprising growing season-long continuous measurements of a corn crop for leaf active fluorescence, canopy SIF, hyperspectral reflectance, and GPP. With partial correlation analysis we evaluated the contributions of iPAR, fAPAR and APAR to the SIF-GPP relationship at the canopy scale. In parallel, we used active fluorescence measurements to investigate the energy partitioning in leaves to reveal the relationship between fluorescence and photosynthesis at the photochemical level.

## 2 Materials and methods

### 2.1 Study site

Field measurements were collected in 2017 at the Optimizing Production inputs for Economic and Environmental Enhancement (OPE[3]) field site (De Lannoy et al., 2006) at the US Department of Agriculture's (USDA) Agricultural Research Service (USDA-ARS) in Beltsville, MD, USA (39.0306° N 76.8454° W, UTC-5). The site is instrumented with a 10 m eddy-covariance tower and a height-adjustable tower (i.e., 1.5-4 m tall) supporting the optical spectral measurements and surrounded by corn (*Zea mays* L.) fields. The two towers were located within the same field that was provided the optimal (100%) nitrogen application for this climate zone, separated by approximately 120 m. Three distinct growth phases of the corn canopy were discerned: Young stage (Y) from DOY 192 to 209, Mature stage (M) from DOY 220 to 235 and Senescent stage (S) from DOY 236 to 264.

### 2.2 Field measurements

The field measurements included active fluorescence observations made on individual leaves, as well as canopy reflectance and SIF retrievals. These were supplemented by carbon fluxes and meteorological data from the site's instrumented tower. These measurements cover the 2017 growing season from day-of-year (DOY) 192 to DOY 264, except for the period from DOY 210 to DOY 219. The main field measurements used in this study are listed in Table 1. In what follows, we briefly introduce the measurements used in the present study (the field campaign was described in detail in Campbell et al., 2019).

*[Insert Table 1 here]*

The site's eddy covariance tower-based system provided 30-minute GPP fluxes continuously collected throughout the growing season. An infrared gas analyzer (Model LI-7200, LI-COR Inc., Lincoln, NE, USA) measured net ecosystem exchange (NEE), which was further partitioned into GPP and ecosystem respiration ($R_e$) using a standard approach (Reichstein et al., 2005)

which extrapolated nighttime values of $R_e$ into daytime values using air temperature measurements.

Canopy spectral measurements were collected by using a field spectroscopy system, the FLoX (JB Hyperspectral Devices UG, Germany), between 7:00 and 20:00 (local time) with a time sampling interval from 1-3 minutes. The system consists of two spectrometers: a QEpro spectrometer (Ocean Optics, Dunedin, FL, USA) and a FLAME-S spectrometer (Ocean Optics,

Dunedin, FL, USA). The QEpro measured down-welling irradiance and up-welling radiance with a 0.3 nm spectral resolution at Full Width at Half Maximum (FWHM) between 650 and 800 nm, which were used to retrieve SIF. The FLAME-S measured the same up-welling and down-welling fluxes but between 400 to 1000 nm with a lower spectral resolution (FWHM of 1.5 nm), which were used to compute canopy values for reflectance ($R$) and to estimate incident PAR ($iPAR_{canopy}$) and $fAPAR_{canopy}$. These TOC measurements were collected from approximately 1.5 m above the canopy at nadir, covering a 25° field of view

(0.66 m diameter at ground level) as reported in (Yang et al., 2020).

Leaf fAPAR ($fAPAR_{leaf}$) was measured on six days spaced across the growing season (n= 18 samples per day). The leaf absorptance spectra between 350 and 2500 nm for nine leaves were measured in the laboratory with an ASD FieldSpec 4 spectrometer (Malvern Panalytical, Longmont, CO, USA) and an ASD halogen light source coupled with an integrating sphere.

The mean $fAPAR_{leaf}$ values per date were computed: $0.92 \pm 0.007$ (i.e., mean $\pm$ stdv) on DOY 192; $0.92 \pm 0.01$ on DOY 199; $0.91 \pm 0.01$ on DOY 221; $0.90 \pm 0.03$ on DOY 222; $0.82 \pm 0.03$ on DOY 240; and $0.75 \pm 0.05$ on DOY 263. Finally, $fAPAR_{leaf}$ on the rest of the days was linearly interpolated/extrapolated from those measurements. Therefore, $fAPAR_{leaf}$ values ranged from 0.93 to 0.70 across the growing season.

Leaf-level active fluorescence measurements were collected by using an automated MoniPAM fluorometer system (Walz, Germany) and five MoniPAM emitter-detector probes, which were operated using a MoniPAM Data Acquisition system (Porcar-Castell et al., 2008). Three probes were positioned to measure sunlit leaves in the upper canopy and the remaining two probes collected measurements on shaded leaves within the lower canopy. The fluorometer collected continuous steady state fluorescence ($F_s$) and maximal fluorescence ($F_m$) every 10 minutes during the day and night. The MoniPAM measured

chlorophyll fluorescence induced by an internal, artificial light source, which produces modulated light with constant intensity (Baker, 2008; Schreiber et al., 1986). In addition to leaf fluorescence measurements, the MoniPAM also measured leaf temperature by an internal temperature sensor and incident PAR ($iPAR_{leaf}$) by a PAR quantum sensor. Leaf APAR ($APAR_{leaf}$) was computed as the product of iPARleaf and $fAPAR_{leaf}$.

### 2.3 Data quality control and sampling

Data quality control of canopy reflectance, SIF and GPP measurements was conducted prior to the analysis. First, measurements collected on 29 rainy or densely clouded days were excluded, because SIF retrieval is generally reliable under clear-sky conditions for which changes are gradual in concert with illumination but not under cloud cover or mostly cloudy conditions when large, unpredictable fluctuations of illumination occur (Chang et al., 2020). Second, a window-based outlier detection was applied to incident PAR data collected by the FLoX to identify unrealistic short-term fluctuations in atmospheric

conditions leading to unreliable SIF retrievals. The fluctuations were detected by finding the iPARcanopy measurements that were not within $\pm$ 3 times the standard deviation for the mean of seven consecutive measurements. Once all cases with fluctuating atmospheric conditions were identified, the reflectance, GPP and SIF measurements acquired within $\pm$half hour of their occurrence were excluded from the analysis. Finally, the remaining FLoX measurements were re-sampled into the 30-minute temporal resolution of the eddy covariance measurements.

### 2.4 Calculation of canopy SIF, fAPAR and APAR

The QEpro spectral measurements were used to compute Top-of-Canopy (TOC) SIF in the $O_2$-A absorption feature at around 760 nm ($F_{760}$). SIF was retrieved using the spectral fitting method (SFM) described in Cogliati et al. (2015). Canopy iPAR (iPAR$_{canopy}$) was computed from the irradiance spectra collected with the FLAME-S spectrometer as the integral of irradiance over the spectral region from 400 to 700 nm. Canopy fAPAR was approximated by using the Rededge NDVI (Normalized

Difference Vegetation Index) (Miao et al., 2018; Viña and Gitelson, 2005):

$$fAPAR = 1.37 \cdot RededgeNDVI - 0.17 \tag{2a},$$

where

$$RededgeNDVI = \frac{R_{750} - R_{705}}{R_{750} + R_{705}} \tag{2b},$$

where reflectance at specific wavelengths is utilized ($R_\lambda$:705 and 750 nm). Rededge NDVI is a widely used index for estimating

fAPAR, and Viña and Gitelson (2005) suggest it as an optimal index for fAPAR among various other vegetation indices in corn canopies. We, however, have tested several other indices for estimating fAPAR, including the enhanced vegetation index (EVI) (Huete et al., 2002; Xiao et al., 2004) and the green NDVI (Viña and Gitelson, 2005), and found that the choice among the three indices had little impact on the results in section 3.1. We also computed the photochemical reflectance index PRI= $\frac{R_{531} - R_{570}}{R_{531} + R_{570}}$ (Gamon et al., 1992), as an indicator of diurnally reversible canopy heat dissipation efficiency $\Phi_{Ncanopy}$.

### 2.5 Quantifying energy partitioning from leaf fluorescence measurements

The continuous MoniPAM measurements offered a way for assessing the dynamics of energy partitioning in photosystem II (PSII). The pathways include photochemistry (P), fluorescence emission (F) and heat dissipation (H). H is further categorized

as a sustained thermal dissipation (D) and a reversible energy-dependent heat dissipation (N). N is controlled by mechanisms that regulate the electron transport of the photosystems and is related to photo-protection mechanisms and NPQ (Baker, 2008).


Relative fluorescence emission efficiency ($\Phi_F^*$) was derived from the MoniPAM steady state fluorescence measurements $F_s$ with a correction for time-varying leaf absorption in the growing season. The correction is needed because $F_s$ responds to the absorbed measurement light rather than the incident measurement light:

$$\Phi_F^* = \frac{F_s}{\text{fAPAR}_{\text{leaf}}} \tag{3}$$


MoniPAM maximal fluorescence measurements ($F_m$), together with the steady state fluorescence ($F_s$), allows the assessment of the absolute efficiencies of absorbed light energy for photochemistry ($\Phi_P$) and the reversible energy-dependent heat dissipation ($\Phi_N$) of PSII. The usual approach to obtain $\Phi_P$ is to 'switch off' photochemistry by applying a saturating light to leaves, so that the fluorescence measurements in the presence and absence of photochemistry ($F_s$ and $F_m$), can be estimated
(Maxwell and Johnson, 2000). A generic expression of $\Phi_P$ proposed by Genty et al. (1989) was used:

$$\Phi_P = 1 - \frac{F_s}{F_m} \tag{4}$$

Unlike photochemistry, it is difficult to fully inhibit heat dissipation. Nevertheless, long duration dark-adaptation can reduce reversible heat dissipation to zero. Then, fluorescence measurements acquired in the presence and absence of reversible heat
dissipation can be estimated. We took the expression proposed by Hendrickson et al. (2004) for $\Phi_N$:

$$\Phi_N = \frac{F_s}{F_m} - \frac{F_s}{F_m^o} \tag{5}$$

where $F_m^o$ is the highest (or maximal) value obtained for dark-adapted leaf fluorescence measurements in the absence of reversible heat dissipation; the pre-dawn value of $F_m$ is typically used as an estimate of true maximal dark-adapted fluorescence (Maxwell and Johnson, 2000). Alternative expressions of $\Phi_N$ can be found in the literature, but they are equivalent and
convertible to each other. For example, Eq. 5 can be rewritten as $\Phi_N = (1 - \Phi_P)(1 - \frac{F_m}{F_m^o})$. Furthermore, it can be expressed as a function of a commonly used fluorescence parameter NPQ, which is defined as $\frac{F_m^o}{F_m} - 1$ (Baker, 2008). In that formulation, $\Phi_N = (1 - \Phi_P)\frac{NPQ}{NPQ+1}$.

The expression of the sum of $\Phi_F$ and $\Phi_D$ (symbolized as $\Phi_{F+D}$) is straightforward, because the sum of the efficiencies of the
four pathways ($\Phi_F$, $\Phi_P$, $\Phi_D$ and $\Phi_N$) is always unity and $\Phi_{F+D} = 1 - \Phi_N - \Phi_P$, and

$$\Phi_{F+D} = \frac{F_s}{F_m^o} \tag{6}$$

Further separation of $\Phi_F$ and $\Phi_D$ from $\Phi_{F+D}$ is difficult, because neither can be inhibited. However, relative efficiency of the sustained heat dissipation ($\Phi_D^*$) across the growing season can be inferred from the pre-dawn values of $F_m$ (i.e., $F_m^o$). Because $F_m^o$ was measured during the night in the absence of both reversible heat dissipation and photochemistry, a change in $F_m^o$ must be caused by a change in the sustained heat dissipation. Therefore, we can take the maximal pre-dawn $\Phi_{F_m}^* = \frac{F_m^o}{\text{fAPAR}_\text{leaf}}$, (when $\Phi_D^*$ is minimal) as a reference and express $\Phi_D^*$ across the growing season as:

$$\Phi_D^* = 1 - \frac{{F_m^o}/{\text{fAPAR}_\text{leaf}}}{\max\limits_{192 \leq \text{DOY} \leq 264}[{F_m^o}/{\text{fAPAR}_\text{leaf}}]} \tag{7}$$

Photosynthetic light use efficiency can be predicted as a function of leaf temperature, ambient radiation levels, intercellular $CO_2$ concentrations $C_i$, and other leaf physiological parameters (e.g., photosynthetic pathways, maximum carboxylation rate at optimum temperature $V_{cmo}$) by using a conventional photosynthesis model of Collatz et al., (1992; 1991). Van der Tol et al., (2014) established empirical relationships between fluorescence emission efficiency and photosynthetic light use efficiency under various environmental conditions by using active fluorescence measurements. With these relationships, the fraction of the absorbed radiation by a leaf emitted as fluorescence and dissipated as heat can be simulated. The MoniPAM system measured leaf temperature and incoming radiation intensity. We reproduced the efficiencies of photochemistry, fluorescence, and reversible and sustained heat dissipation by using the biochemical model of Van der Tol et al., (2014). The two most influential model input variables, leaf temperature and incoming radiation, were measured by using the MoniPAM. $V_{cmo}$ was set to 30 μmol m$^{-2}$ s$^{-1}$ at 25 °C, a recommended value for the corn crop (Houborg et al., 2013; Wullschleger, 1993; Zhang et al., 2014). The rest of the model variables (e.g., $C_i$) to their default values. In this way, we simulated the efficiencies for the temporal resolution of the MoniPAM measurements (i.e., 10 minutes) and examined the relationship among the efficiencies as predicted by the biochemical model.

## 2.6 Statistical analysis

Pearson correlation coefficients ($\rho$) were computed to evaluate the relationships between pairs of observations, such as $\Phi_P$ and $\Phi_F^*$, or GPP and SIF. In addition to the correlation coefficients, partial correlation coefficients were computed to measure the degree of association between GPP and SIF, where the effect of a set of controlling variables was removed, including fAPAR, iPAR and APAR. Partial correlation is a commonly used measure for assessing the bivariate correlation of two quantitative variables after eliminating the influence of one or more other variables (Baba et al., 2004). The partial correlation between x and y given a controlling single variable z was computed as

$$\rho_{x,y(z)} = \frac{\rho_{x,y} - \rho_{x,z}\rho_{y,z}}{\sqrt{1-\rho_{x,z}^2}\sqrt{1-\rho_{y,z}^2}} \tag{8}$$

where $\rho_{x,y}$ is the Pearson correlation coefficient between x and y. Note that the relationships reported in this study are statistically significant (p-value<0.01) unless otherwise stated.

Partial correlation can be calculated to any arbitrary order. $\rho_{x,y(z)}$ is a first-order partial correlation coefficient, because it is conditioned solely on one variable (z). We used a similar equation to calculate the second-order partial coefficient that accounts for the correlation between the variables x and y after eliminating the effects of two variables z and q (de la Fuente et al., 2004).

$$\rho_{x,y(zq)} = \frac{\rho_{x,y(z)} - \rho_{x,q(z)}\rho_{y,q(z)}}{\sqrt{1-\rho_{x,q(z)}^2}\sqrt{1-\rho_{y,q(z)}^2}} \tag{9}$$

## 3 Results

### 3.1 Relationship between canopy SIF and GPP observations

Fig. 1a confirms the linear SIF-GPP relationship reported in previous studies and shows that $F_{760}$ and GPP were strongly correlated with an overall correlation $\rho = 0.83$. This correlation was slightly stronger than the relationship between $APAR_{canopy}$ and GPP (an overall $\rho = 0.80$, Fig. 1b). The $APAR_{canopy}$-GPP relationship was apparently comprised of parallel groups of responses (colors) with large variation in GPP exhibited for the same levels of $APAR_{canopy}$ (Fig. 1b). This relationship complies with the common understanding of the response of photosynthesis to light showing the well-known saturation with irradiance as photosynthesis of the whole canopy gradually shifts from light limitation to carbon limitation, while the unexplained (by light intensity) variation in GPP can be attributed to stomatal aperture responses and a time-varying carboxylation capacity, especially in the upper sunlit canopy, which experienced larger variations of light intensity. SIF, which is affected by both light and carbon limitations, shows a more linear response to GPP than $APAR_{canopy}$ (Figs. 1a vs. 1b).

*[Insert Figure 1 here]*

Incoming radiation (i.e., $iPAR_{canopy}$) had a strong, positive linear relationship with SIF, GPP and $APAR_{canopy}$ (as shown in Figs. 1 and 2). We investigated these canopy-scale relationships with partial correlation analysis as diagrammed in Fig. 2, where for simplicity's sake, the subscripts denoting "canopy" variables were omitted in the diagram. Our team (Yang et al., 2020) and others (Miao et al., 2018; Migliavacca et al., 2017) have previously demonstrated that in addition to incoming radiation intensities, the energy available for photochemistry and fluorescence (i.e., $APAR_{canopy}$) is strongly affected by canopy structure and leaf biochemistry. As a result, there were cases of low SIF, GPP and/or $APAR_{canopy}$ values at high $iPAR_{canopy}$ (Fig. 1, red and orange dots), and *vice versa* high SIF, GPP and/or $APAR_{canopy}$ values at low $iPAR_{canopy}$ (Fig. 1, blue and violet dots). This is shown in the correlation diagram as well (Fig. 2) which indicates that SIF, GPP and $APAR_{canopy}$ were all moderately dependent on leaf biochemistry as well as on canopy structure according to their correlations with $fAPAR_{canopy}$, i.e., $\rho_{SIF,fAPAR}=$ 0.60, $\rho_{GPP,fAPAR}= 0.58$ and $\rho_{APAR,fAPAR}= 0.70$ (i.e., numbers in bold, blue text, Fig. 2). Compared with either $iPAR_{canopy}$ or $fAPAR_{canopy}$, $APAR_{canopy}$ as their product (located in center, Fig. 2) can better explain the variations in SIF and GPP observations, with Pearson correlations of $\rho = 0.92$ and 0.80, respectively.

*[Insert Figure 2 here]*

After removing the effects of this important controlling variable that affects both SIF and GPP, namely APAR$_{canopy}$, the correlation between GPP and SIF was weak ($\rho_{SIF,GPP(APAR)}$= 0.27; refer to results below the triangle's baseline). In contrast, the correlation between SIF and GPP remained significant after controlling for the effects of the components of canopy APAR, either fAPAR$_{canopy}$ or iPAR$_{canopy}$, i.e., $\rho_{SIF,GPP(fAPAR)}$= 0.72, $\rho_{SIF,GPP(iPAR)}$= 0.66 (equations below the triangle, Fig. 2).

We further investigated how the SIF-GPP relationship varied seasonally with growth stage and diurnally with time of the day (Fig. 3). The SIF-GPP correlation was significantly lower (by 22-27%) for the senescent canopy than for the young and mature canopy. The Pearson correlation coefficient was highest when the canopy was fully developed with the underlying surface covered in the mature stage ($\rho$ = 0.77, Fig. 3b). As for the different times of a day, we found that their correlations were the

strongest in the afternoon ($\rho$ = 0.89) while $\rho$ was only 0.76 when the data were acquired in the morning (Figs 3d vs. 3f).

*[Insert Figure 3 here]*

### 3.2 Dynamics of energy partitioning in photosystems

The continuously acquired active fluorescence measurements offered a way to assess the dynamics of energy partitioning in

photosystems and facilitated the understanding of the relationship between fluorescence and photosynthesis before aggregation to the canopy, at the photochemical level. We investigated how the partitioning evolved over time.

During the nighttime, as can be seen from the responses in the dark-bars in Fig. 4, the photosystem energy partitioning was stable for all leaves through the night, whether they were designated as sunlit or shaded during the day. Three efficiencies ($\Phi_P$,

$\Phi_F^*$ and $\Phi_D^*$) showed little overnight change, and the reversible heat dissipation $\Phi_N$ was always close to zero. This null response for $\Phi_N$ agrees with the known status/behavior of the most important driver of reversible heat dissipation, the xanthophyll pigment cycle, which reverts overnight to the energy-neutral form violaxanthin, and then converts during the day to antheraxanthin in moderately high light levels and subsequently to zeaxanthin at high light levels by chemical de-epoxidation (Middleton et al., 2016; Müller et al., 2001).

*[Insert Figure 4 here]*

During the daytime, there were dramatic day-to-day changes in energy partitioning to photochemistry, fluorescence and reversible heat dissipation (Fig. 4). Generally, both $\Phi_F^*$ and $\Phi_N$ increased during mornings to midday and decreased afterwards, except that $\Phi_N$ exhibited unexplained midday dips during the senescent stage. On the other hand, $\Phi_P$ decreased

during mornings to midday lows and increased afterwards (i.e., $\Phi_P$ diurnals were bowl-shaped, as shown in many studies).

The changes in $\Phi_N$ and $\Phi_P$ corresponded closely with the changes in incident radiation, while $\Phi_F^*$ changes corresponded closely with the dynamics in incident radiation in the morning but not at midday when the radiation level was high. The light levels influenced the partitioning of absorbed radiation into the three different pathways. However, other factors, such as leaf temperature, intercellular $CO_2$ concertation, and $V_{cmax}$ (which varied seasonally) also played roles in determining the absolute efficiencies of each pathway.

At the seasonal scale (Fig. 4), however, the nighttime energy partitioning over the three other pathways ($\Phi_P$, $\Phi_F^*$ and $\Phi_D^*$) displayed substantial variations. The nighttime $\Phi_P$ was about 0.82 on all days during the young and mature stages, which is close to the theoretical maximal value (Zhu et al., 2008), but it was only about 0.64 during the senescent stage. Similarly, the nighttime relative light use efficiency of fluorescence $\Phi_F^*$ clearly decreased as the canopy development progressed from the physiologically robust (young and mature) stages to the senescent stage. For example, the nighttime $\Phi_F^*$ for both the sunlit and shaded leaves was above 60 in the young stage but was around 50 in the senescent stage. The seasonal/growth stage decreases during nighttime in both $\Phi_F^*$ and $\Phi_P$ were attributed to an increase of sustained heat dissipation $\Phi_D^*$ since nighttime $\Phi_N$ was always close to zero. In extrapolating $\Phi_D^*$ to daytime, we assumed that the sustained heat dissipation remained unchanged within any full day (from 0:00 to 24:00), but noticeable changes in $\Phi_D^*$ sometimes occurred between two consecutive days, e.g., between $\Phi_D^*$ on DOY 194 and DOY 195, and between DOY 230 and DOY 231, as indicated in Fig. 4.

Although the sunlit and shaded leaves had similar seasonal and diurnal patterns, some interesting differences are observed. As expected, the radiation levels were higher for the sunlit leaves than for the shaded leaves, which produced higher $\Phi_F^*$ for the sunlit leaves and slightly lower $\Phi_P$ at the young and mature stages. In comparison to the difference in $\Phi_F^*$, the difference in $\Phi_P$ was less pronounced. At the senescent stage $\Phi_P$ of the shaded leaves was substantially lower than sunlit leaves despite receiving lower radiation, which normally would lead to higher $\Phi_P$. This could be attributed to the different leaf ages and functionality of sunlit and shaded leaves; for example, shaded corn leaves senesce earlier than sunlit leaves. Additionally, $\Phi_D^*$ of sunlit leaves was higher than the shaded leaves while $\Phi_N$ of the sunlit and shaded leaves was similar.

It is evident that the contribution to the photosynthetic process by the combined nighttime fluorescence and sustained heat dissipation group ($\Phi_{F+D}$, red color in Fig. 5) increased through the growing season, to competitively reduce photochemical efficiency ($\Phi_P$, green color), especially during senescence. The increase of sustained heat dissipation (Fig. 4) also resulted in a decrease of $\Phi_P$ in the daytime as the young and mature stages progressed through the senescent stage, although $\Phi_P$ can vary substantially during the daytime. Additionally, the diurnally reversible heat dissipation ($\Phi_N$, gold color) was generally higher at the senescent stage than at the young and mature stages, which contributed to the reduction in photochemical efficiency as well. In the pie charts, we focus on the energy partitioning in both nighttime and midday since they portray the potential

maximal $\Phi_P$ (i.e., the photosynthetic reaction centers in the nighttime are mostly open) and the steady-state $\Phi_P$ at the most common time of day for satellite observations, respectively.

*[Insert Figure 5 here]*

The pie charts (Fig. 5) clearly show how the partitioning of the relative efficiency pathway contributions changed with growth stage on the three representative clear sky days. The nighttime $\Phi_P$ was reduced by 20% between the young and senescent stages, while $\Phi_{F+D}$ increased by 19% during senescence. The pie charts also clearly show the very strong role of reversible

heat dissipation in limiting midday photosynthesis throughout the growing season. For example, the per cent contribution for the pathways from the young crop (DOY 196) was 35% for $\Phi_P$, 23% for $\Phi_N$, and 42% for $\Phi_{F+D}$. The corresponding values for leaves in the mature crop (DOY 232) were 31%, 14%, and 56%. And for the leaves in the senescing crop (DOY 254), the corresponding values were 14%, 26%, and 61%. Combining these together, Fig. 5 further highlights the complexity of energy efficiency dynamics underlying the photosynthetic process.

**3.3 Relationships among photosynthesis, fluorescence and heat dissipation at leaf level**

Next, we examine the leaf-level efficiency terms obtained from *in situ* measurements, in terms of their combined responses. The first set compares $\Phi_F^*$ and $\Phi_P$, in the context of variable iPAR$_{\text{leaf}}$ (Figs. 6a, b). This figure clearly shows that the relationship between $\Phi_F^*$ and $\Phi_P$ during daylight (9:00 - 17:00) was different for the sunlit (sun adapted) vs. shaded (shade adapted) leaves, since the sunlit leaves were more often exposed to iPAR above 1000 μmol m$^{-2}$s$^{-1}$. The higher $\Phi_P$ values were

obtained for relatively low iPAR$_{\text{leaf}}$, whether sunlit or shaded. For sunlit leaves, $\Phi_F^*$ and $\Phi_P$ were positively correlated overall ($\rho = 0.53$, Fig. 6a) and in conditions with moderate to high light intensity (iPAR$_{\text{leaf}}$ >500 μmol m$^{-2}$ s$^{-1}$, excluding blue and teal colored dots), $\rho = 0.60$. In contrast, at low light intensity (iPAR$_{\text{leaf}}$ <500 μmol m$^{-2}$ s$^{-1}$, blue dots), correlation between $\Phi_F^*$ and $\Phi_P$ was weak and negative for $\Phi_P$>0.4. These two efficiency terms were uncorrelated in shaded leaves (Fig. 6b), and $\Phi_F^*$ was much lower in the shaded than in sunlit leaves. The correlations on individual days are presented in Fig. 8a, which shows that

positive correlations between $\Phi_F^*$ and $\Phi_P$ are more often for sunlit leaves than shaded leaves.

*[Insert Figure 6 here]*

At the seasonal scale, the midday $\Phi_F^*$ and $\Phi_P$ values (the average of all values acquired between 11:00 and 14:00) had a quasi-linear, positive relationship for both the sunlit and shaded leaves when iPAR$_{\text{leaf}}$ >500 μmol m$^{-2}$s$^{-1}$ (Fig. 6c). In contrast, at low

average midday light intensities, the relationships were clearly negative. The $\Phi_P$ values tended to decrease with the increasing light intensities while the relationship between $\Phi_F^*$ and iPAR$_{\text{leaf}}$ was not definite. However, the ranges for $\Phi_F^*$ in sunlit and shaded leaves clearly represent two populations: $\Phi_F^*$ shaded was < 110 (Fig. 6c) whereas $\Phi_F^*$ sunlit > 100 (Fig. 6c). These results could have implications for interpreting canopy-scale measurements.

The linear relationship obtained between $\Phi_P$ and $\Phi_N$ was considerably stronger for both sunlit and shaded leaves (Figs. 7a, b) than the correlation between $\Phi_F^*$ and $\Phi_P$ previously shown for sunlit leaves (Fig. 6a). Here, both sunlit and shaded leaves showed consistent and strong linear decreases in $\Phi_P$ as $\Phi_N$ increased (Figs. 7a, b) in response to increase in the intensity of incoming light (iPAR$_{leaf}$, Fig. 4). Furthermore, the $\Phi_P$ and $\Phi_N$ relationships definitely varied in response to the sustained heat dissipation ($\Phi_D^*$, levels represented in the color bar) in a similar fashion for both sunlit and shaded leaves, although higher $\Phi_D^*$

values (orange and red dots) were obtained in sunlit leaves. The efficiency of photochemistry obviously declined at higher $\Phi_D^*$, as indicated with the arrows in Fig. 7, especially pronounced in sunlit leaves. For shaded leaves, there were cases with higher $\Phi_D^*$ that did not result in lower $\Phi_P$ (the orange dots in Fig. 7b). When both thermal dissipations were fully manifested, the $\Phi_P$ was greatly reduced; in sunlit leaves, this reduction was ~40%. The correlations on individual days are presented in Fig. 8b, which shows $\Phi_N$ and $\Phi_P$ are negatively correlated with each other for both sunlit and shaded leaves.

*[Insert Figure 7 here]*
   *[Insert Figure 8 here]*

   At the seasonal scale, as can be seen from Figs. 4 and 5, $\Phi_P$ decreased while $\Phi_D^*$ increased as the canopy progressed through its growth stages. Their seasonal relationship is depicted in Fig. 7c, showing a same-day comparison of the midday $\Phi_P$ value

(the average between 11:00 and 14:00), as a function of $\Phi_N$ across the growing season noting that $\Phi_D^*$ remained unchanged within any full day. Generally, $\Phi_N$ and $\Phi_P$ exhibited an overall negative correlation, but clearly their relationship was regulated by $\Phi_D$. This is seen in the different midday $\Phi_P$ responses at high vs. low $\Phi_D^*$ values. At the same level of $\Phi_N$ (around 0.05), the magnitudes of midday $\Phi_P$ varied by up to 0.45 (65%, from 0.37 to 0.61 in Fig. 7c) due to variations in the efficiency of the sustained heat dissipation which varied between 0.1 and 0.6.


   We have shown that $\Phi_P$ was regulated by heat dissipation (Figs. 5 and 7), and was moderately correlated with $\Phi_F^*$ at light levels above 500 $\mu$mol m$^{-2}$ s$^{-1}$ but was negatively correlated $\Phi_F^*$ at lower light levels (Fig. 6). With the dynamics of energy partitioning within the photosystem now quantified, we interpret the emerging relationship between photochemical and fluorescence efficiencies, namely $\Phi_P$ and $\Phi_F^*$ (Table 2), in the context of thermal dissipation efficiencies ($\Phi_N$, $\Phi_D^*$). After

eliminating the effects of both sustained and reversible heat dissipation, $\Phi_P$ and $\Phi_F^*$ were negatively and equally correlated ($\rho$ = -0.75) for both sunlit and shaded leaves. As surprising as this is, the presence of either sustained or reversible heat dissipations changed this underlying negative relationship ($\Phi_P$ vs. $\Phi_F^*$) into an observed apparent positive relationship at leaf scale, which contributes to the positive relationship of GPP and SIF at canopy scale. In fact, accounting for the effects of either $\Phi_N$ or $\Phi_D^*$ reduced the correlation coefficients between $\Phi_P$ and $\Phi_F^*$. For sunlit leaves, controlling for only $\Phi_N$ reduced the correlation

from 0.53 to 0.05 (by ~0.48 units); after controlling for only $\Phi_D^*$, the correlation dropped by 0.45 units to 0.08. For shaded leaves this reduction was from 0.10 to -0.31 after controlling for $\Phi_N$, or to -0.35 after controlling for $\Phi_D^*$. The reduction of the correlation between $\Phi_P$ and $\Phi_F^*$ were caused by diurnal variations in $\Phi_N$ and seasonal variations in both $\Phi_N$ and $\Phi_D^*$.

Results of model simulations are presented in Figs 9 and 10.  In comparison with Figs. 6 and 7 that describe our *in situ* measurements, these two figures show that the biochemical model outputs were more successful in describing photosynthetic efficiency as a function of reversible heat dissipation ($\Phi_N$) than fluorescence efficiency ($\Phi_F$).  Specifically, for the $\Phi_P$-$\Phi_F$ relationships, the Fig. 9 simulation shows some similarity to the Fig. 6 measurements, but clearly does not capture the different responses we obtained for sunlit versus shaded leaves. However, Fig. 10 does generally replicate the general responses

expected based on in situ measurements (Fig. 7), portraying the strong negative impact of $\Phi_N$ on $\Phi_P$, but it doesn't convey the variability captured under field conditions. These differences occurred in the simulations because we did not consider the physiological (i.e., enzyme activity) or physical (i.e., thickness, pigment ratios) differences among leaves at different growth stages.  Neither did we consider the physical differences or photochemical potential differences (e.g., total chlorophyll content and Chl a/b ratios; rubisco activity) between sunlit and shaded leaves in this modelling experiment. Therefore, it is to be

expected that the simulations for sunlit and shaded leaves would be similar, and not displaying the differences observed in field measurements. Furthermore, we did not include changes in leaf display geometry induced by low water stress (i.e., drought) in the simulations, but it is a common phenomenon in corn plants in the field. Another likely reason contributing to the differences between simulations and observations is that in using the model of Van der Tol et al. (2014) to derive $\Phi_F$ from $\Phi_P$, $\Phi_D$ is assumed to be a constant and $\Phi_N$ is empirically estimated as a function of $\Phi_P/\Phi_{P0}$. The observations shown in

Figs. 4 and 5 prove that $\Phi_D$ varied over the growing season, and therefore, cannot be considered as a constant. These findings may help improve the modelling of $\Phi_F$ at the biochemical level and thus improve our understanding of the relationship between SIF and GPP at the canopy scale.


### 3.4 Comparison of light use efficiencies at leaf and canopy levels

The responses of the efficiencies to APAR and the relationships between these efficiencies are diagrammed in Fig. 11, showing the Pearson correlation coefficients between pairs of variables, for leaves (Fig. 11a) that were either sunlit or shaded (indicated in bold, blue text), and for canopy (Fig. 11b).


At the leaf level, we see that $\Phi_F^*$ showed moderate correlation to $\Phi_P$ for sunlit leaves ($\rho = 0.53$) but very low correlation to $\Phi_P$ for shaded leaves ($\rho = 0.10$). The highest correlations were negative, denoting inverse relationships between $\Phi_N$ and $\Phi_P$ (-0.74 sunlit and -0.87 shaded), whereas similar positive correlations (0.64 sunlit and 0.68 shaded) were obtained between

$\Phi_N$ and $APAR_{leaf}$ (located in center, Fig. 11a), as expected since $\Phi_N$ is well known to be light-level sensitive when invoking

the xanthophyll cycle. Notice that all of the high correlations (>0.64 or <-0.74), whether positive or negative, are located on the left-hand side of Fig. 11a, which compares efficiencies of photochemistry with efficiencies of reversible thermal dissipation ($\Phi_N$) and their connection through $APAR_{leaf}$. The remaining correlations on the right-hand side, between $\Phi_F^*$ and either $\Phi_P$, $\Phi_N$, or $APAR_{leaf}$, are significantly lower (from -0.33 to 0.53).


At the canopy level, $\Phi_{Fcanopy}$ also showed moderate correlation to $\Phi_{Pcanopy}$ with $\rho = 0.37$ (Fig. 11b, for the scatter plot between $\Phi_{Pcanopy}$ and $\Phi_{Fcanopy}$, see Fig. A1), which falls between the values for sunlit and shaded leaves (Fig. 11a). An inverse relationship between $\Phi_{Pcanopy}$ and $APAR_{canopy}$ (-0.41) was found at the canopy level, but this correlation was much weaker than that at the leaf level (-0.75 for both sunlit and shaded leaves). The photochemical reflectance index PRI= $\frac{R_{531}-R_{570}}{R_{531}+R_{570}}$

(Gamon et al., 1992), as an indicator of $\Phi_{Ncanopy}$, appeared to have no correlations with either $APAR_{canopy}$ or $\Phi_{Pcanopy}$, while at the leaf level these three variables had strong correlations (located on the left-hand side of Fig. 11a). Comparing the efficiencies obtained from the leaf- and canopy-level measurements (i.e., $\Phi_{Pcanopy}$ vs. $\Phi_P$ or $\Phi_{Fcanopy}$ vs. $\Phi_F^*$), no clear relationships were found ($\rho <0.1$, data are shown in Fig. A2).

*[Insert Figure 12 here]*
*[Insert Table 3 here]*

Comparison of Fig 11a with Fig. 12a reveals that the strength of correlations between pairs of variables describing energy partitioning for both sunlit and shaded leaves increased for most pairs when evaluated at midday vs. diurnal measurements
(Table 3). For example, three pairs showed notable correlation enhancements for sunlit leaves in midday across the growing season: the negative correlations between $\Phi_N$ and $\Phi_F^*$ (from -0.33 to -0.45) and between $APAR_{leaf}$ and $\Phi_F^*$ (from -0.10 to -0.27), and the positive correlation between $\Phi_P$ and $\Phi_F^*$ (from 0.53 to 0.62). Shaded leaves showed similar but even stronger responses than sunlit leaves overall at midday, and especially for these same three pairs: $\Phi_N$ vs. $\Phi_F^*$ (shaded, from -0.23 to -0.45), and $\Phi_N$ vs. $\Phi_F^*$ (from 0.10 to 0.27). In addition, for shaded leaves, the midday positive correlation between $APAR_{leaf}$
and $\Phi_N$ also was higher (from 0.68 to 0.77) as was the negative correlation between $\Phi_N$ and $\Phi_P$ (from -0.87 to -0.92), while the positive correlation between $APAR_{leaf}$ and $\Phi_F^*$ became a weak negative association (from 0.25 to -0.14). No noticeable correlation changes occurred for sunlit leaves at midday vs. daily measurements for these two pairs: $\Phi_N - \Phi_P$ ($\rho \approx$ -0.75) or $APAR_{leaf} - \Phi_N$ ($\rho \approx$ 0.61). The negative correlations were equal for sunlit and shaded leaves between $\Phi_N$ and $\Phi_P$ whether determined for daily or at midday, but the midday correlation was stronger (from -0.75 to -0.81). Especially noteworthy are
the strong negative correlations that were observed (Table 3) in sunlit and shaded leaves for $\Phi_N$ and $\Phi_P$ (between -0.74 and -0.92) and $APAR_{leaf}$ and $\Phi_P$ (between -0.75 and -0.81).

Comparison of Fig. 11b and Fig. 12b reveals that at the canopy scale all correlations between variable pairs were relatively modest (e.g., $\rho \leq \pm 0.55$) but were higher at midday than for daily observations across the growing season, except for $\Phi_{Ncanopy}$ (as estimated with the PRI) vs. $\Phi_{Fcanopy}$ ($\leq -0.07$, indicating no relationship). For the remaining five pairs, the strongest and most improved responses at midday were between $\Phi_{Pcanopy}$ and $\Phi_{Fcanopy}$ (from 0.37 to 0.53) and between APAR$_{canopy}$ and $\Phi_{Pcanopy}$ (from -0.41 to -0.55), with a stronger association also seen for APAR$_{canopy}$ vs. $\Phi_{Fcanopy}$ (from -0.25 to -0.32). It is apparent that the canopy responses based on remote sensing, without including critical information on the sunlit/shaded canopy illumination fractions (Figs 11b, 12b), were less successful in describing the energy partitioning that was provided at the leaf level (Figs. 11a, 12a).

## 4 Discussion

### 4.1 Physical basis for the SIF-GPP relationship

Incoming radiation intensity, leaf biochemistry, leaf and canopy structure all affect APAR$_{canopy}$, the energy source for photosynthesis, SIF and heat dissipation. We found an equal contribution of iPAR$_{canopy}$ and fAPAR$_{canopy}$ to the observed SIF-GPP canopy relationship. The correlation coefficients between SIF and GPP remained relatively high after controlling either term. In stark contrast, after holding APAR (their product, iPAR$_{canopy}$ x fAPAR$_{canopy}$) constant, the SIF-GPP canopy correlation coefficient was reduced from 0.83 to 0.27. This demonstrates the dominance of APAR$_{canopy}$ in determining the relationship between SIF and GPP. Compared to APAR$_{canopy}$, SIF was slightly better correlated with GPP (Fig. 1). The physiological information implied in GPP was seemingly better expressed with SIF than APAR$_{canopy}$.

The interfering effects of $f_{esc}$ at canopy scale have not been considered explicitly. They are implicit in the relations of $\rho_{SIF,GPP(APAR)}$ (Qiu et al., 2019). When accounted for, they may provide a better estimate of the correlation attributable to the physiological response of photosystems (i.e., $\rho_{SIF,GPP(APAR,fesc)} > 0.27$). The magnitude and sign of $\rho_{SIF,GPP(APAR)}$ are nevertheless consistent with the moderate correlation we found between leaf $\Phi_F^*$ and $\Phi_P$ for sunlit leaves and the weak correlation for shaded leaves (Figs. 6 and 11a). In addition, we found that the positive relationship between $\Phi_F^*$ and $\Phi_P$ at the seasonal time scale is dominated by the progressive increase of sustained heat dissipation ($\Phi_D^*$) during senescence. In contrast, there was significant diurnal but no clear seasonal variation of $\Phi_N$.

### 4.2 Physiological basis for the SIF-GPP relationship

Clear differences between the responses of sunlit and shaded leaves influence the correlation for the canopy as a whole. The $\Phi_F$ and $\Phi_P$ of sunlit leaves exposed to moderate or high iPAR$_{canopy}$ exhibited a moderately strong linear relationship ($\rho = 0.53$), while no such relationship existed for leaves at low iPAR$_{canopy}$ (independent of whether the leaves were classified as sunlit or shaded leaves). Leaves regularly receiving sunlight during development (sunlit leaves) differ structurally and biochemically

from leaves in lower light positions in the canopy. Shaded leaves are often thinner, smoother, and larger in surface area (Dai et al., 2004). The larger shaded leaves provide a larger area for absorbing light energy for photosynthesis where light levels are lower. In contrast, smaller sunlit leaves provide less surface area for the loss of water through transpiration which is higher due to direct exposure to solar radiation. The greater mesophyll thickness of sunlit leaves produces more inter-cellular spaces to facilitate increased carbon dioxide conductance into their smaller chloroplasts, producing greater rates of photosynthesis per unit leaf area in sunlit leaves (Givnish, 1988; Jackson, 1967).

The investigated crop has a C4 photosynthetic pathway, in which dark and light reactions are separated, and carboxylation takes place under a high CO2 concentration. This strongly suppresses photorespiration in C4 vegetation, resulting in a higher water use efficiency and lower sensitivity to heat and high vapour pressure deficit than for C3 vegetation. Liu et al. (2017) reported that the GPP–SIF relationship was much stronger for a C4 crop (corn) than a C3 crop (wheat). They showed that while $\Phi_{Fcanopy}$ of the C3 and C4 crops were similar, the $\Phi_{Pcanopy}$ of corn was much higher than for wheat. Because of different photosynthetic pathway and the contribution of photorespiration, the SIF-GPP relationship of C3 vegetation is more complicated in the corn crop examined in this study.

Compared to the relationship between leaf fluorescence emission efficiency, total heat dissipation (both D and N) provided a robust and direct indicator of leaf photosynthetic light use efficiency (Fig. 7). In particular, the variation of reversible heat dissipation better explains the diurnal variation of leaf photosynthetic light use efficiency, whereas the sustained heat dissipation contributes to the seasonal variation. Reversible heat dissipation is the main regulating mechanism for the dissipation of absorbed photosynthetically active radiation energy (Adams et al., 1989; Demmig-Adams et al., 1996; Heber et al., 2006; Huang et al., 2006). Our study confirms its dominant role for the corn crop with field measurements and finds that the reversible heat dissipation is responsible for the positive relationship between $\Phi_F$ and $\Phi_P$ of sunlit leaves at diurnal scales, though less so at seasonal scales when sustained heat dissipation is dominant (Fig. 6). Remote sensing monitoring at the canopy/landscape scale of the reversible efficiency of heat dissipation is still challenging. It is well known that changes in $\Phi_N$ are often associated with changes in leaf green reflectance due to changes in the de-epoxidation state (DEPS) of xanthophyll cycle pigments. The photochemical reflectance index (PRI) utilized the link between the biochemical changes within xanthophyll cycle expressed with a narrow-band green reflectance, providing a way to remotely assess photosynthetic light use efficiency (Gamon et al., 1992; Garbulsky et al., 2011), but the link becomes partially obscured at canopy scale due to the effects of canopy structure and sun-observer geometry (Hilker et al., 2009; Middleton et al., 2009). Because of these interfering effects, canopy PRI showed very weak overall relationship with APAR$_{canopy}$ ($\rho$=0.28, Fig. 11b), which clearly differed from the connection between $\Phi_N$ and APAR$_{leaf}$ at the leaf level ($\rho \geq 0.64$, Fig. 11a).

Since the reversible heat dissipation pathway is such a strong competitor to photochemistry, especially in the sunlit canopy fraction, it seems very important to fully understand the green reflectance link to the energy regulation via the xanthophyll

cycle, and then develop radiative transfer modelling approaches to translate this link to the canopy level. In this regard, Vilfan et al. (2018) extended the Fluspect leaf radiative transfer model to simulate xanthophyll driven leaf reflectance dynamics. Further efforts on implementing this extended model in canopy radiative transfer models will connect efficiencies of photochemistry and reversible heat dissipation to canopy reflectance observations. This may open new opportunities to estimate photosynthetic light use efficiency and improve GPP estimation using remote sensing methods *in situ* and from space.

### 4.3 Physically and physiologically joint effects on the SIF-GPP relationship

The canopy equivalent efficiencies ($\Phi_{Fcanopy}$ and $\Phi_{Pcanopy}$) are composed of integrals of the efficiencies of leaves of the sunlit and shaded canopy fractions. The correlation between the canopy effective equivalents of $\Phi_F$ and $\Phi_P$ may be expected to take a value between the equivalent correlation of leaf-level $\Phi_F$ and $\Phi_P$ for sunlit leaves ($\rho = 0.53$) and for shaded leaves ($\rho = 0.10$). This means that the ability to view the SIF and reflectance hot spots (whether they occur together or not) from sunlit leaves varies with viewing angle and time of day (e.g., illumination angle, diffuse light). We suggest that these factors strongly affect $f_{esc}$. Therefore, they must, in turn, affect the success of remote sensing relationships for SIF-GPP (Yang and Van der Tol, 2018). Likewise, these factors also affect the variability of the APAR-GPP relationship (Dechant et al., 2020; Qiu et al., 2019), and the relationship of photosynthetic light use efficiencies at leaf and canopy levels (i.e., $\Phi_P$ and $\Phi_{Pcanopy}$) (e.g., Middleton et al., 2019). However, it is worth noting that active fluorescence measurements are spectrally integrated signals, whereas canopy passive SIF observations are obtained at one wavelength. As a result, the leaf-level fluorescence emission and photosynthetic light use efficiencies derived from active fluorescence measurements differ spectrally from the canopy-level efficiencies ($\Phi_{Fcanopy}$ and $\Phi_{Pcanopy}$). This difference may also play a role in upscaling leaf-level to canopy-level relationship between $\Phi_F$ and $\Phi_P$.

The exact correlation between $\Phi_{Fcanopy}$ and $\Phi_{Pcanopy}$ at canopy scales depends on both the relative contributions of sunlit and shaded leaves to the canopy equivalents and the native correlation of the efficiencies at leaf level (Köhler et al., 2018; Mohammed et al., 2019). Canopy structure dictates the relative abundance and thus the relative weights of these contributing factors to the canopy equivalent $\Phi_F$ and $\Phi_P$. The weight is not only determined by leaf class abundance, but also by the relative magnitude of the SIF and GPP response of the leaf classes. Sunlit leaves during daytime usually constitute a greater contribution to the effective canopy efficiencies than shaded leaves, simply because sunlit leaves tend to emit a higher SIF signal and, at the same time, produce a higher GPP. This suggests that the correlation between the canopy effective equivalents of $\Phi_F$ and $\Phi_P$ tends to be closer to the correlations of leaf-level $\Phi_F$ and $\Phi_P$ for sunlit leaves ($\rho =0.53$) than for shaded leaves.

The LUE models as shown in Eq. 1 are, essentially, one-big-leaf models. The one-big-leaf approach assumes that canopy photosynthesis or SIF have the same relative responses to the environment as any single leaf, and that the scaling from leaf to canopy is therefore linear (Friend, 2001). However, sunlit and shaded leaves clearly showed a different $\Phi_F$-$\Phi_P$ relationship

(Figs. 6 and 11). In order to better interpret the SIF-GPP relationship, we recommend a revision of the LUE model of SIF and

GPP (Eq. 1) by separating the contributions of sunlit and shaded leaves:

$$GPP = \sum_{n=sunlit,shaded} iPAR \cdot fAPAR^n \cdot \Phi_P^n \qquad (10a),$$

$$SIF = \sum_{n=sunlit,shaded} iPAR \cdot fAPAR^n \cdot \Phi_F^n \cdot f_{esc}^n \qquad (10b),$$

This approach updates the existing one-big-leaf LUE models into two-leaf (or two-big-leaf) LUE models. The idea of

differentiating sunlit and shaded leaves in vegetation modelling has been applied in predicting canopy temperature and

photosynthesis, and an improved ability of PRI to track canopy light use efficiency was shown when including both sunlit and

shaded leaves in model simulations of field results (Dai et al., 2004; Luo et al., 2018; Wang and Leuning, 1998; Zhang et al.,

2017). Qiu et al, (2019) incorporated a fluorescence simulation in the Boreal Ecosystem Productivity Simulator (BEPS, Liu et

al., 1997), which is a two-leaf process-based model. More classes of leaves with varying ambient temperatures and incident

radiation levels can be examined using more explicit models, such as SCOPE (Soil-Canopy-Observation of Photosynthesis

and Energy fluxes, Van Der Tol et al., 2009), BETHY-SCOPE (the Biosphere Energy Transfer Hydrology model coupled with

SCOPE, Norton et al., 2018) or DART (the Discrete Anisotropic Radiative Transfer model, Gastellu-Etchegorry et al., 2017).

Although the concept of differentiating sunlit and shaded leaves is implemented in these model, the functional variation of the

two categories of leaves is not considered. Moreover, the role of sunlit fraction in explaining SIF-GPP relationship has not

been explored. The two-leaf LUE models consider the major difference of leaves in a canopy, and are relatively simpler

compared with SCOPE and DART (Parazoo et al., 2020) but more realistic compared with one-big-leaf LUE models in linking

SIF and GPP.

The fraction of sunlit canopy is determined by canopy structure and the direction of incoming light as well as the fraction of

diffuse light. Hence, it is expected that these factors will affect the contribution of sunlit and shaded leaves to the canopy SIF-

GPP correlation. Furthermore, the instantaneous sun-view angle geometry affects where the sunlit leaves occur during the day

and the likelihood of their being viewed at particular angles (e.g., nadir). This means that the ability to view the SIF hot spot

emitted from sunlit leaves varies with viewing angle and time of day. We suggest that these factors strongly affect $f_{esc}$ which

must, in turn, affect the SIF-GPP remote sensing relationship (Yang and Van der Tol, 2018).

Intuitively, in fully contiguous vegetation canopies the leaves in the upper layer (which are often sunlit) contribute a major

fraction to the whole canopy of APAR, whereas fAPAR$_{shaded}$ is small. Therefore, compared with the efficiencies of shaded

leaves, $\Phi_F^{sunlit}$ and $\Phi_P^{sunlit}$ have much larger relative contributions to $\Phi_{Fcanopy}$ and $\Phi_{Pcanopy}$, respectively. Hence, a stronger

relationship between SIF and GPP for dense canopies is expected since $\Phi_F^{sunlit}$ and $\Phi_P^{sunlit}$ are more tightly connected than

$\Phi_F^{shaded}$ and $\Phi_P^{shaded}$. For dense canopies, the leaves in the upper layer absorb a large fraction of incoming radiation, and less

radiation can penetrate to the lower layers and be absorbed by shaded leaves. This results in that the quantity

fAPAR$_{sunlit}$/fAPAR$_{total}$ is generally higher for dense canopies, such that the contribution to canopy SIF and GPP from sunlit leaves is higher for dense canopies than for sparse canopies. This insight can provide some explanation for the seasonally varying results describing canopy SIF and GPP (Fig. 3 a-c), where the SIF-GPP relationship varied with the growth stages: for the Young crop ($\rho = 0.72$); Mature crop ($\rho = 0.77$); and the Senescent crop ($\rho = 0.50$).

Furthermore, the effects of diffuse light (the diffuse/direct iPAR ratio) on the relationship between SIF and GPP can be explained by the revised equation (Eq. 10). When the fraction of diffuse light is higher (e.g., a hazy, or cloudy conditions), there is greater iPAR penetration into lower canopy layers (the shaded leaves). As a result, fAPAR$_{shaded}$ increases while fAPAR$_{sunlit}$ decreases. This leads to a higher contribution of shaded leaves to the SIF-GPP relationship at canopy level, and weakens the SIF-GPP correlation. This was indeed observed in earlier field measurements reported in Miao et al. (2018), which showed that both the SIF-GPP correlation and the correlation between the SIF/APAR and GPP/APAR ratios were significantly weaker under cloudy conditions than sunny conditions. We excluded the data collected on rainy or densely clouded days in the analysis to ensure the quality of SIF retrieval. Nevertheless, the relative fraction of diffuse light is also a possible cause for the diurnally varying correlation between SIF and GPP (Fig. 3 d-f), where the SIF-GPP relationship varied at different times of day: for the data acquired in the morning ($\rho = 0.76$); for the data acquired in the midday ($\rho = 0.83$); and for the data acquired in the afternoon ($\rho = 0.89$). This highlights the unique physiological information of SIF for monitoring GPP, and the joint effects of incoming radiation, canopy structure and leaf physiology on the SIF-GPP relationship. We suggest that the canopy structure, illumination and viewing conditions, and especially the foliage thermal dissipation must be taken into account to accurately represent the physiological underpinnings of the observed SIF-GPP relationships.

A simple model was used to examine the sensitivity of the fraction of sunlit canopy to LAI, leaf angle distribution function (LIDF) and solar zenith angles ($\theta_s$). Considering a vegetation canopy as a turbid medium consisting of leaves, the instantaneous sunlit fraction can be estimated as a function of the direction of incoming light, canopy LAI ($L$) and leaf angle distribution. In stochastic models describing the transfer of radiation in plant canopies, the probability of the leaves being sunlit at a specified vertical height $x$ (i.e., x= 0 referring to top of canopy, x= -1 referring to bottom of canopy) can be estimated as $P_s(x) = \exp(kLx)$, where $L$ is canopy LAI and k the extinction coefficient, which is determined by the solar direction and leaf angle distribution (He et al., 2017; Stenberg and Manninen, 2015). The computation of $k$ is explicitly given in Verhoef (1984) by projecting the leaf area into the direction of the sun. In the model SCOPE (Van Der Tol et al., 2009), the total fraction of sunlit canopy LAI is the integral of $P_s$ in the vertical direction given as:

$$P_s = \frac{1}{kL}(1 - \exp(-kL)) \tag{11}$$

The effects of LAI, leaf angle distribution function (LIDF) and solar zenith angles ($\theta_s$) on the instantaneous sunlit canopy fraction are presented in Fig. 13. In line with our intuitive understanding, the fraction of sunlit canopy decreases with increasing

canopy LAI in denser canopies. This fraction also decreases with increasing solar zenith angle, which are also affected by the leaf angle distribution. The important quantity for our purposes is the relative (not absolute) angular difference between the sun and leaf positions. Eq. 11 gives the prediction for the total fraction of sunlit canopy, but the fraction of sunlit LAI at a given height and thus the vertical variation of $P_{sun}$ can be predicted in a similar way. The calculation of the fraction of sunlit canopy LAI shown in Eq. 11 is based on a turbid assumption of the vegetation canopy. Corn has a simple canopy architecture and a corn canopy can be considered as turbid medium. However, for forests or other more complex canopies, other structural characteristics, e.g., the clumping of foliage (Liu et al., 1997; Qiu et al., 2019), affect the gap probability of a vegetation canopy layer and the associated light penetration, and should be considered when separating sunlit and shaded leaves in the canopy.

*[Insert Figure 13 here]*

A limitation of the current SCOPE capability for describing physiological responses is related to capturing the changing light environments that affect estimates of the sunlit/shaded fractions. This is because SCOPE and most radiative transfer models for vegetation assume steady state conditions and lack temporal memory of state variables at different times. SCOPE predicts the sunlit/shaded fractions at one moment while the shaded and sunlit leaves discussed in this paper are a result of long-term adaption to the light conditions (i.e., sun-adapted and shade-adapted leaves). Nevertheless, we can gain insights into relationships under specified conditions, which can serve as new information to be used in updating the models. A possible way is to predict the light distribution inside the canopy with varying sun positions (e.g., a diurnal cycle). In this way, sun-adapted and shade-adapted leaves can be differentiated according to the probability of being illuminated for a longer period instead of for a single moment in time. A leaf is sun-adapted when it is almost always illuminated at various sun positions or different time in a day. In contrast, a shade-adapted leaf is rarely or occasionally illuminated for various sun positions. Furthermore, different physiological traits of sun-adapted and shade-adapted can be taken into account in the model.

## 4.4 Combined use of TOC reflectance and SIF for GPP estimation

SIF observed at the top of a canopy is a fraction of total emitted SIF by all the leaves in the canopy due to the reabsorption and scattering effects. In section 4.1, we inferred that the correction of TOC SIF for $f_{esc}$ can result in a better correlation to GPP, and in section 4.3 we discussed the difference between leaf- and canopy-level efficiencies caused by the canopy structural and sun-observer geometry. Apart from separating sunlit and shaded leaves in the LUE models proposed in section 4.3, employing corrections to SIF for interfering structural and angular effects are possible ways to enhance understanding of the relationship between SIF and GPP.

Several studies have been conducted to convert TOC SIF to total emitted SIF by the canopy (SIF_{tot}) for a better estimation of GPP (e.g., Lu et al., 2020; Qiu et al., 2019). A direct way to estimate $f_{esc}$ or SIF_{tot} is by using a radiative transfer model (e.g., SCOPE and DART), but this approach requires leaf and canopy characteristics to drive the models and has obvious limitations

in applications. Because TOC reflectance and TOC SIF are similarly determined by leaf biochemistry, canopy structure and sun-observer geometry, we can use TOC reflectance to explain vegetation biochemical and structural, and bidirectional effects on TOC (Yang et al., 2019, 2020; Yang and Van der Tol, 2018). This can be achieved by retrieving required leaf and canopy characteristics for running the radiative transfer model from TOC reflectance (Yang et al., 2019). Alternatively, we can establish a direct link between TOC reflectance and $f_{esc}$ skipping the retrieval of vegetation properties by inverting a radiative transfer model. This can be achieved by exploring the similarity of radiative transfer of intercepted incident light and emitted SIF. We established such a link, which states that the ratio of far-red reflectance ($R$) to the product of canopy interceptance ($i_0$) and leaf albedo ($\omega$) is an accurate estimate of canopy scattering of far-red SIF (i.e., $f_{esc} = R/i_0\omega$) (Yang and Van der Tol, 2018). Furthermore, we found that the product of $f_{esc}$ and fAPAR$_{canopy}$ can be well approximated by a reflectance index, which is called fluorescence correction vegetation index (FCVI) and is given as the difference of near-infrared (NIR) and broadband visible (VIS) reflectance acquired under identical sun-canopy-observer geometry of the SIF measurements (i.e., FCVI $= R_{nir} - R_{\overline{vis}} \approx f_{esc} \times$ fAPAR$_{canopy}$) (Yang et al., 2020). With the above mentioned link and index, it is possible to estimate $f_{esc}$ and canopy total emitted SIF irradiance at 760 nm $F_{760}^{tot}$ (i.e., $F_{760}^{tot} = \pi$ iPAR $\cdot$ fAPAR $\cdot \Phi_{Fcanopy}$) by correcting radiance of the TOC SIF in the viewing direction ($F_{760}$) for the escape probability

$$f_{esc} = \text{FCVI/fAPAR}_{canopy} \tag{12}$$

$$F_{760}^{tot} = \pi F_{760}/f_{esc} \tag{13}$$

*[Insert Figure 14 here]*

We estimated $F_{760}^{tot}$ using Eqs. 12 and 13 and found that $F_{760}^{tot}$ is not better correlated with GPP compared with $F_{760}$ as indicated by the similar correlation coefficients and RMSEs (Fig. 1a vs Fig. 14). For $F_{760}^{tot}$ and GPP, the Pearson correlation coefficient was 0.82 and RMSE was 0.29 mg m$^{-2}$ s$^{-1}$, while the values were 0.83 and 0.28 mg m$^{-2}$ s$^{-1}$ for $F_{760}$ and GPP. The reason is likely to be the uncertainties in the $f_{esc}$ estimation. The accuracy of $f_{esc}$ estimation with FCVI is largely determined by fAPAR$_{canopy}$, which is difficult to accurately estimate from TOC reflectance alone. In most studies including the present study, fAPAR$_{canopy}$ is usually estimated by using vegetation indices and the accuracy is not always guaranteed. Because SIF is a weak signal, the uncertainties in fAPAR$_{canopy}$ estimation may have a considerable impact on estimating $f_{esc}$ and $F_{760}^{tot}$. Similar problems also exist when using the NIRv (near infrared vegetation index, NDVI×R$_{nir}$) to correct TOC SIF for $f_{esc}$, since fAPAR$_{canopy}$ is required (i.e., $f_{esc} = $ NIRv/fAPAR$_{canopy}$ ) (Zeng et al., 2019). Nevertheless, Lu et al. (2020) found that canopy GPP was bettered correlated with $F_{760}^{tot}$ than with $F_{760}$. Instead of fAPAR$_{canopy}$ and either FCVI or NIRv, they used the original link we established (Yang and Van der Tol, 2018) between TOC far-red reflectance and $f_{esc}$ when estimating $f_{esc}$ (i.e., $f_{esc} = R/i_0\omega$). The important variables $i_0$ and $\omega$ for applying this link were estimated by using field measurements of leaf and canopy characteristics (e.g., leaf chlorophyll contenta and LAI). The study of Lu et al. (2020) not only confirms that canopy total emitted SIF is a better estimate of GPP than TOC SIF, but also supports the importance of fAPAR$_{canopy}$ in estimating $f_{esc}$

when using either NIRv or FCVI. We, therefore, recommend that canopy interceptance $i_0$ be included into measurement protocols in future field campaigns to better monitor GPP based on SIF remote sensing retrievals.

## 5 Conclusions

We have used a unique dataset to explore the relationship between fluorescence and photosynthesis at leaf and canopy levels over a growing season in a corn canopy. We have quantified the contribution of incoming radiation, canopy structure and plant physiology to the SIF-GPP relationship by using partial correlation analysis.

We demonstrate that the observed positive relationship between SIF and GPP is largely due to the fact that both of them are dependent on APAR (i.e., not on iPAR). Incoming radiation and canopy structure had comparable contributions to the SIF-GPP relationship. After eliminating the effects of variable APAR on the SIF-GPP relationship, the apparent positive relationship between SIF and GPP became much weaker. However, there is still some remaining connection due to the functional link between fluorescence and photosynthesis at the leaf level, which is confirmed by active fluorescence measurements.

We also confirm that heat dissipation is responsible for the positive relationship between the efficiencies of fluorescence and photochemistry. Sustained (i.e., diurnally stable) heat dissipation increased through the crop's growth into the senescent stage, which caused the late season decrease in photosynthetic light use efficiency. The seasonal variation in sustained heat dissipation contributed to a moderate positive relationship between the efficiencies of fluorescence and photochemistry at the seasonal scale. At the diurnal scale, the reversible heat dissipation is responsible for the change of photosynthetic light use efficiency.

We propose to use a two-big-leaf LUE model instead of the commonly used one-big-leaf LUE model for interpreting the SIF-GPP relationship. This is because of clearly different relationships between fluorescence emission and photochemical light use efficiencies for sunlit and shaded leaves. The use of the two-big-leaf LUE model leads to a better understanding of the SIF-GPP relationship and its responses to weather conditions, such as clouds and fraction of diffuse light, as well as its responses to canopy structure, such as canopy openness and growth stages. We also suggest to include measurements of canopy interceptance or fAPAR in future field campaigns to allow estimating canopy total emitted SIF from TOC SIF for a better estimation of GPP. This study unravels the individual effects of incoming light, vegetation structure and leaf physiology and highlights their joint effects on the relationship between canopy fluorescence and photosynthesis. Our findings on the physical and physiological basis for the SIF and GPP relationship at the leaf level should, therefore, lead to more mature physiological-physical SIF retrieval approaches, upgrading the current empirical or statistical methods, to facilitate canopy monitoring of photosynthesis from space by based on SIF.

## Appendix A

*[Insert Figure A1 here]*
*[Insert Figure A2 here]*

*Author contributions:* P.Y., E.M., C.vdT and P.C. designed and performed research; P.Y. analyzed the data and prepared the original draft; P.Y., E.M., C.vdT and P.C. reviewed and edited the paper.

*Data availability:* The data is provided as a supplement.

*Competing interests:* The authors declare no conflict of interest.

### Acknowledgements

This work was supported by the Netherlands Organization for Scientific Research, grant ALWGO.2017.018. The collection of field data and the work of co-authors Campbell and Middleton were supported by NASA's Terrestrial Ecology program grant 80NSSC19M0110, Land Cover Land Use Change grant 80NSSC18K0337, and the Biospherc-Sciences Laboratory at NASA Goddard Space Flight Center.

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

 **Tables and figures**

**Table 1: Summary of the main canopy and leaf field measurements used in the analyses.**

|  | Variable | Description | Measuring system | Unit | Temporal resolution |
|---|---|---|---|---|---|
| Canopy | GPP | gross primary production | eddy covariance system | mg m$^{-2}$ s$^{-1}$ | 30 minutes |
|  | $F_{760}$ | canopy SIF at 760nm | QEpro (in FLOX) | mW m$^{-2}$ s$^{-1}$ | 1-3 minutes |
|  | iPAR$_{canopy}$ | TOC incoming PAR | FLAME-S (in FLOX) | $\mu$mol m$^{-2}$ s$^{-1}$ | 1-3 minutes |
|  | fAPAR$_{canopy}$ | canopy fraction of absorbed PAR | FLAME-S (in FLOX) | - | 1-3 minutes |
| Leaf | iPAR$_{leaf}$ | leaf incoming PAR | MoniPAM system | $\mu$mol m$^{-2}$ s$^{-1}$ | 10 minutes |
|  | fAPAR$_{leaf}$ | leaf fAPAR | ASD spectrometer | - | - |
|  | $F_m$ | maximal fluorescence levels | MoniPAM system | - | 10 minutes |
|  | $F_s$ | steady-state fluorescence levels | MoniPAM system | - | 10 minutes |

**Table 2: Correlation coefficients (the first row) and partial correlation coefficients (i.e. controlling for or eliminating separate effects) between fluorescence and photosynthesis. The coefficients are placed in italics if the relationship is not significant (p-value>0.10).**

| $\Phi_F^*$ vs. $\Phi_P$ | Sunlit leaves | Shaded leaves |
|---|---|---|
| Without controls | 0.53 | 0.10 |
| Controlling $\Phi_N$ | *0.05* | -0.31 |
| Controlling $\Phi_D$ | *0.08* | -0.35 |
| Controlling both $\Phi_N$ and $\Phi_D$ | -0.75 | -0.75 |

**Table 3. Correlations between variables describing energy partitioning at leaf and canopy scales. The coefficients are placed in italics if the relationship is not significant (p-value>0.10).**

| Scale | Time | Types | $\Phi_N$ vs. $\Phi_F$ | $\Phi_P$ vs. $\Phi_F$ | $\Phi_N$ vs. $\Phi_P$ | APAR vs. $\Phi_F$ | APAR vs. $\Phi_N$ | APAR vs. $\Phi_P$ |
|---|---|---|---|---|---|---|---|---|
| Leaf | All | Sunlit | -0.33 | 0.53 | -0.74 | -0.10 | 0.64 | -0.75 |
| | | Shaded | -0.23 | 0.10 | -0.87 | 0.25 | 0.68 | -0.75 |
| | Midday | Sunlit | -0.45 | 0.62 | -0.76 | -0.27 | 0.60 | -0.81 |
| | | Shaded | -0.45 | 0.27 | -0.92 | *-0.14* | 0.77 | -0.81 |
| Canopy | All | | *-0.04* | 0.37 | -0.16 | -0.25 | 0.28 | -0.41 |
| | Midday | | *-0.07* | 0.53 | -0.25 | -0.32 | 0.41 | -0.55 |

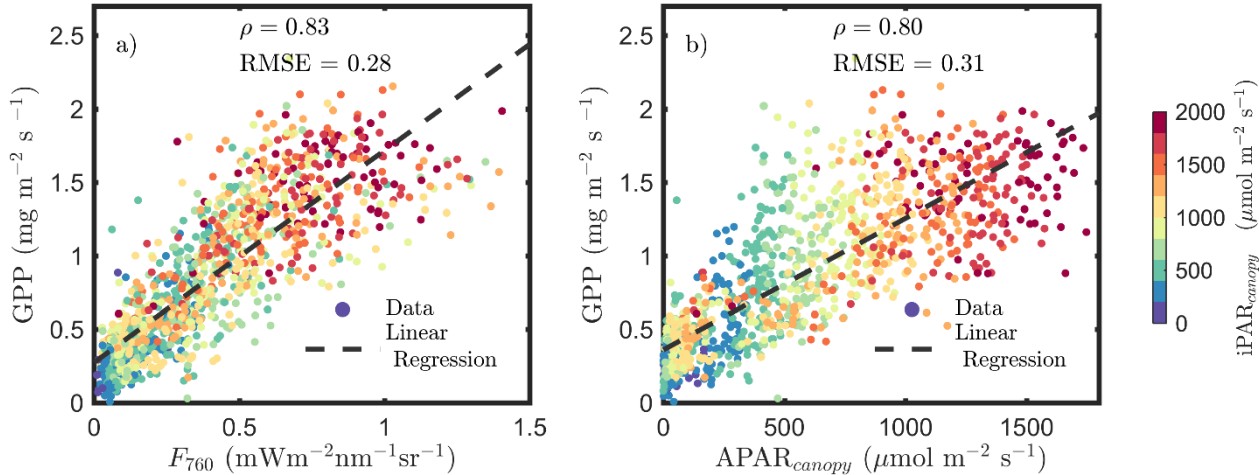

**Figure 1: Relationships between far-red SIF ($F_{760}$) and GPP, and between APAR_canopy and GPP of a corn canopy in the 2017 growing season with half-hour temporal resolution during daylight hours. $F_{760}$ and APAR_canopy were retrieved from FLoX canopy measurements. GPP was obtained from the site's flux tower measurements.**

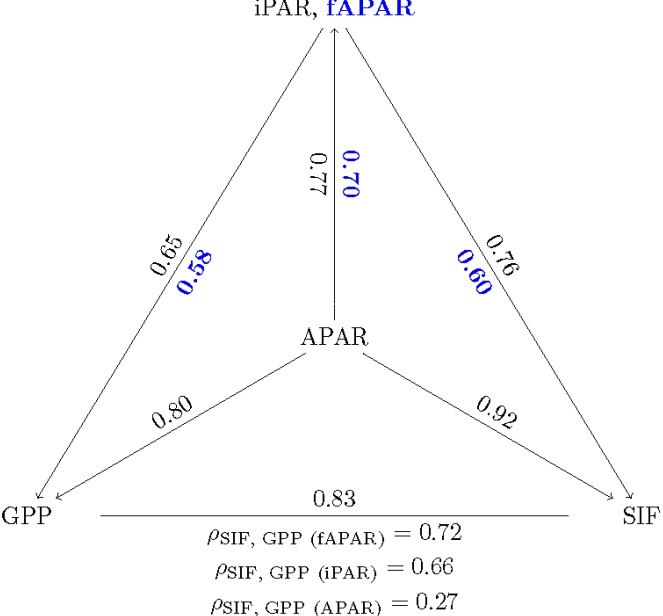

**Figure 2: Pearson correlation coefficients among the canopy variables iPAR_canopy, APAR_canopy, fAPAR_canopy (indicated in bold, blue text), GPP, and SIF for a corn canopy across the 2017 growing season, based on the dataset shown in Fig. 1 (a, b). The partial correlation coefficients between SIF and GPP (listed at the base of the triangle) were determined by removing the effects of the controlling variables, fAPAR, iPAR and APAR, respectively. Measurements had a half-hour resolution.**

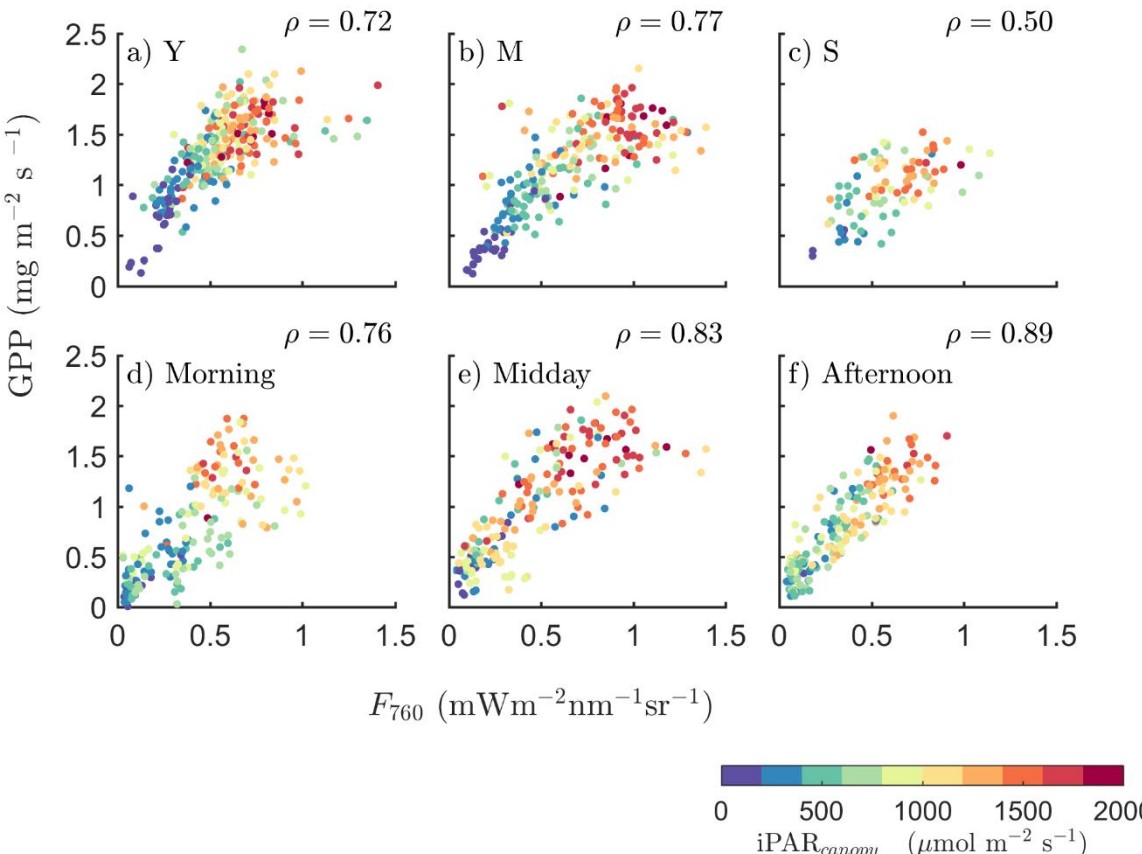

**Figure 3: Relationships between far-red SIF ($F_{760}$) and GPP of a corn canopy across the 2017 growing season with half-hour temporal resolution during daytime hours for three growth stages (a-c):young (Y), mature (M) and senescent (S); for three times of a day (d-f): morning (9:00-11:00), midday (11:00-14:00) and afternoon (14:00-17:00). Colors refer to the iPAR$_{canopy}$ values obtained in conjunction with the GPP and SIF observations, as shown in the legend bar.**

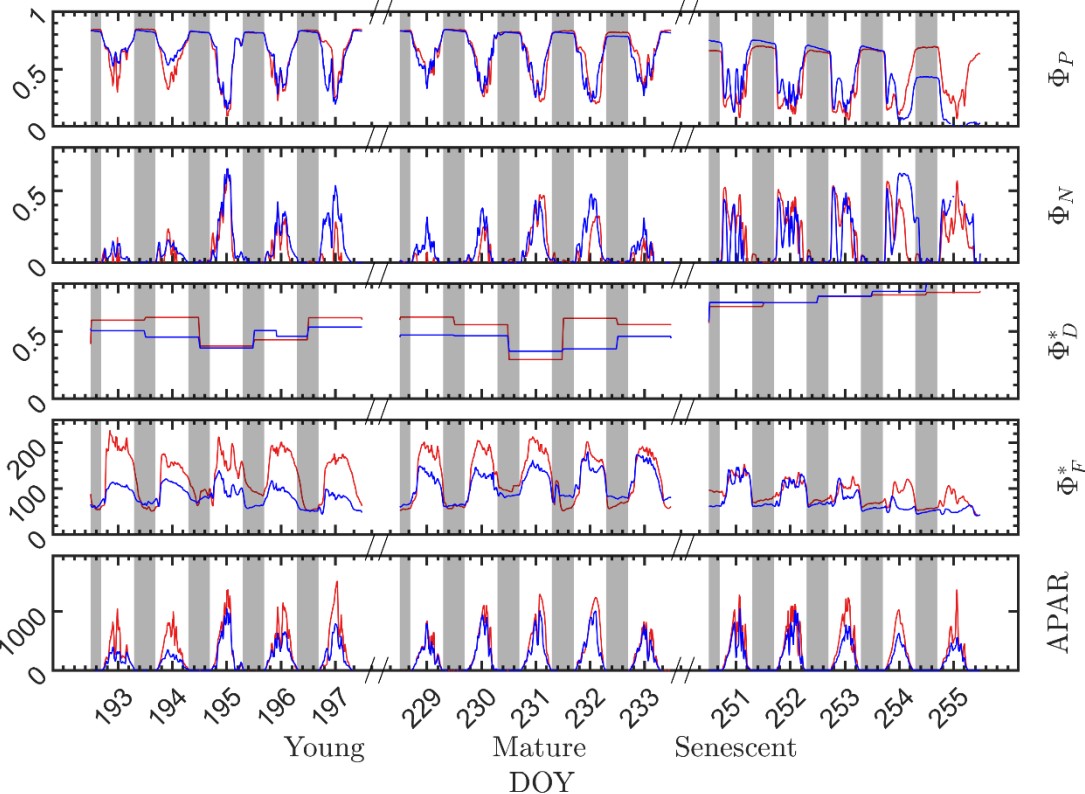

**Figure 4: Photosystem energy partitioning obtained from *in situ* active fluorescence measurements made on individual leaves of a corn canopy during the 2017 growing season. Shown are the absolute light use efficiency of photochemistry ($\Phi_P$), the reversible heat dissipation ($\Phi_N$), the relative light use efficiency of sustained heat dissipation ($\Phi_D^*$), the relative light use efficiency of fluorescence ($\Phi_F^*$) and the photosynthetically active radiation absorbed by individually leaves (APAR$_{leaf}$, µmol m$^{-2}$ s$^{-1}$) for sunlit leaves (red lines) and shaded leaves (blue lines). The nighttime periods from sunset to sunrise of the next day are marked with grey rectangles.**

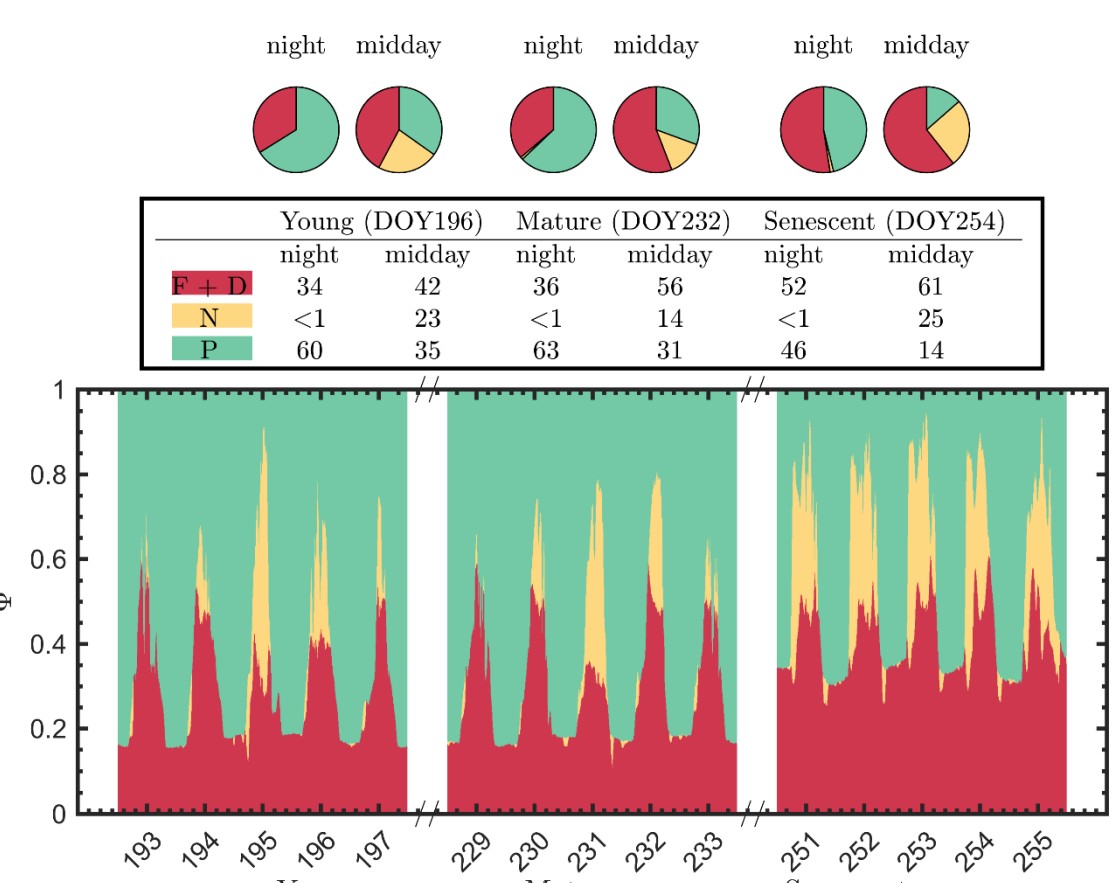

| | Young (DOY196) | | Mature (DOY232) | | Senescent (DOY254) | |
|---|---|---|---|---|---|---|
| | night | midday | night | midday | night | midday |
| F + D | 34 | 42 | 36 | 56 | 52 | 61 |
| N | <1 | 23 | <1 | 14 | <1 | 25 |
| P | 60 | 35 | 63 | 31 | 46 | 14 |

**Figure 5: Summary chart of the efficiency responses presented in Fig. 4 for sunlit leaves. The energy partitioning in both nighttime (sunset - sunrise) and midday (11:00 - 14:00) measurements for one representative date per growth stage (Y, DOY 196; M, DOY 232; and S, DOY 254) is diagrammed in the pie charts. Clearly, the photosynthetic efficiencies (P, green) are constrained, especially during daytime, by the combined action of reversible thermal dissipation efficiency (N, gold) and the F + D (fluorescence and sustained thermal dissipation, red) efficiency.**

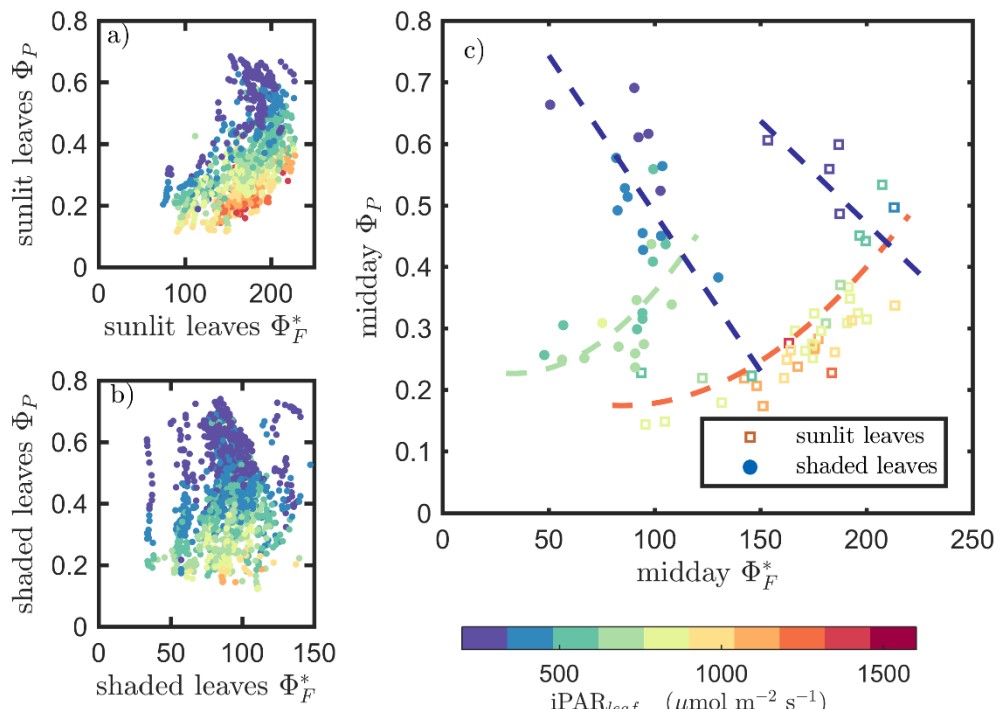

**Figure 6: Relationships between the light use efficiency of photochemistry ($\Phi_P$) and the relative fluorescence light emission efficiency ($\Phi_F^*$) for sunlit leaves and shaded leaves across the 2017 growing season in a corn canopy are shown: all daytime measurements (9:00 - 17:00, a and b); and midday (11:00 - 14:00) seasonally-averaged measurements (c). Colors refer to the iPAR_{leaf} values shown in the legend bar.. The data in (c) were classified into two groups by iPAR_{leaf} with a threshold value of 500 µmol m$^{-2}$ s$^{-1}$.**

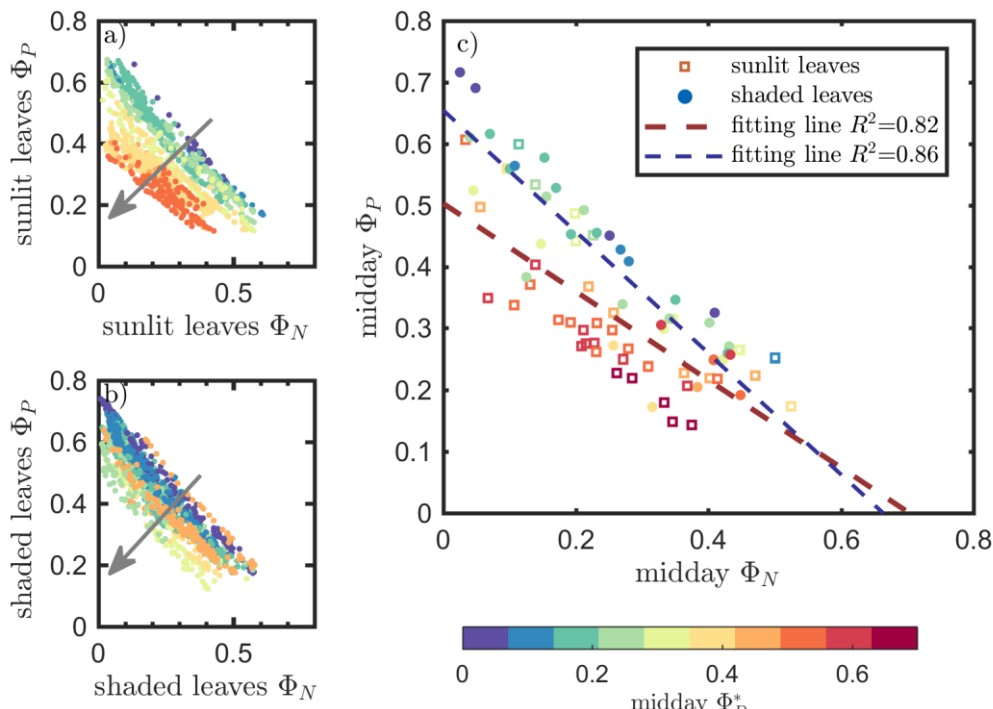

**Figure 7: Relationships between the light use efficiency of photochemistry ($\Phi_P$) and the reversible heat dissipation ($\Phi_N$) for sunlit leaves and shaded leaves across the 2017 growing season in a corn canopy are shown: all daytime measurements (9:00 - 17:00, a and b); and midday (11:00 - 14:00) seasonally-averaged measurements (c). Colors refer to the midday $\Phi*_D$ values shown in the legend bar. . The arrows indicate the shift in linear response between $\Phi_P$ and $\Phi_N$ as $\Phi_D^*$ becomes the dominant energy pathway, thus lowering the photosynthetic potential.**

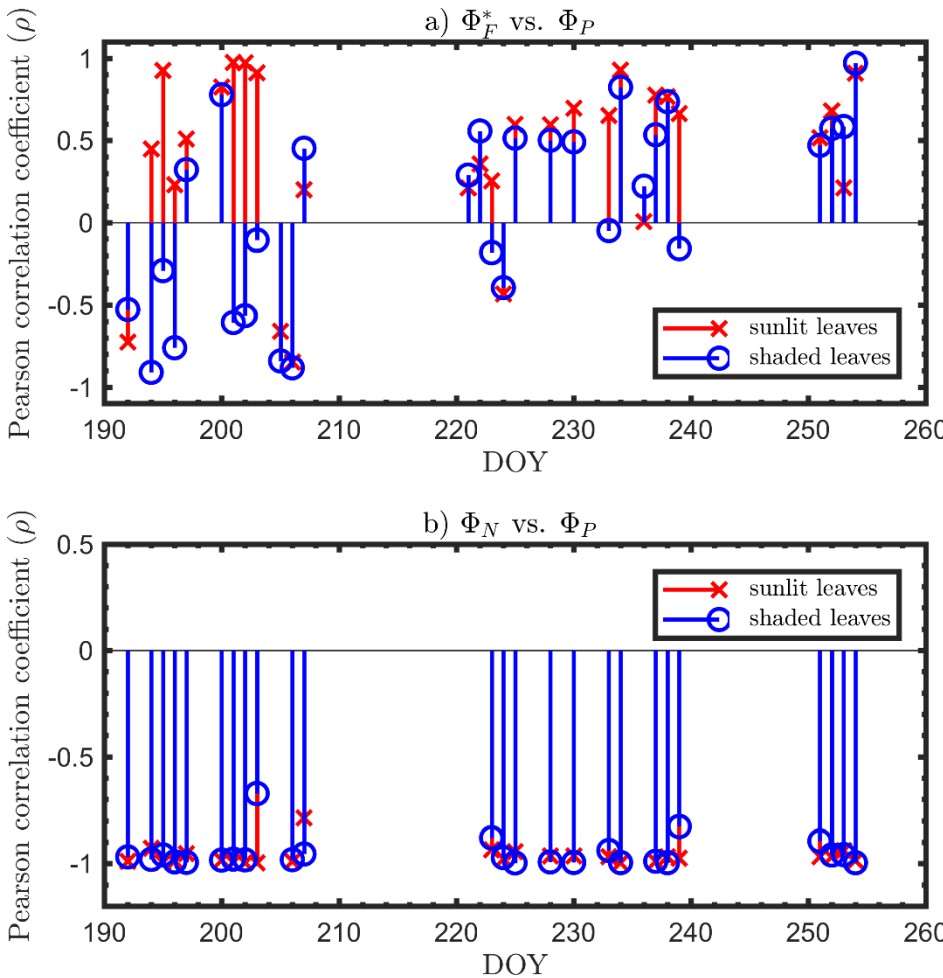

**Figure 8: Diurnal correlations between $\Phi_F^*$ and $\Phi_P$, and between $\Phi_N$ and $\Phi_P$ for sunlit and shaded leaves. The Pearson correlation coefficients for the days with more than five available observations are presented.**

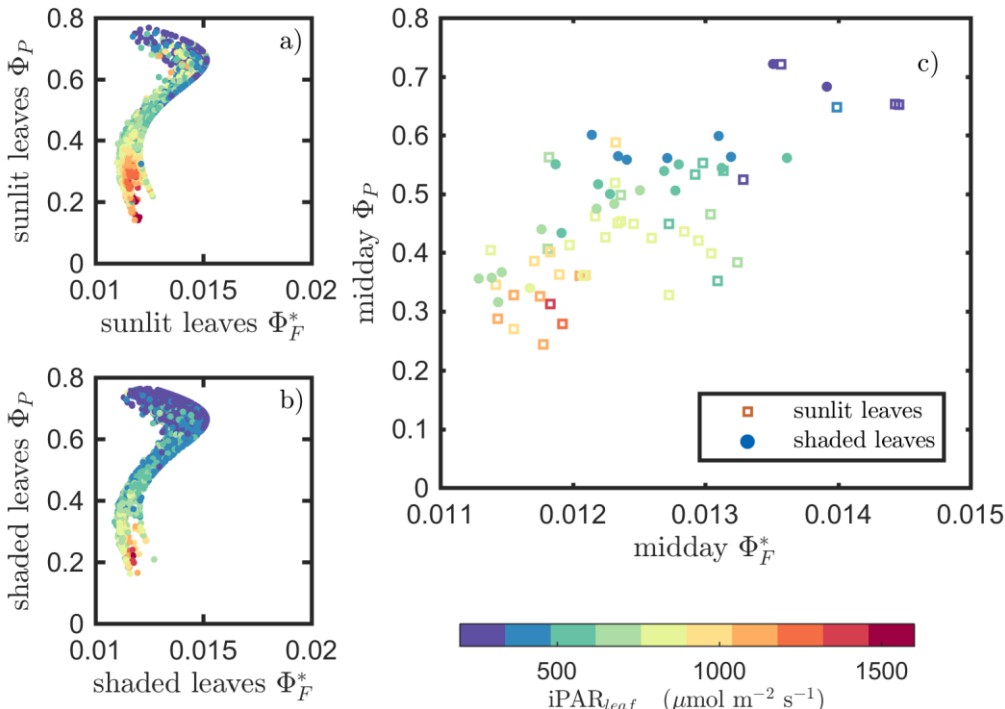

**Figure 9: Reproduction of Fig. 6 with simulated variables from the biochemical model of Van der Tol et al. (2014).**

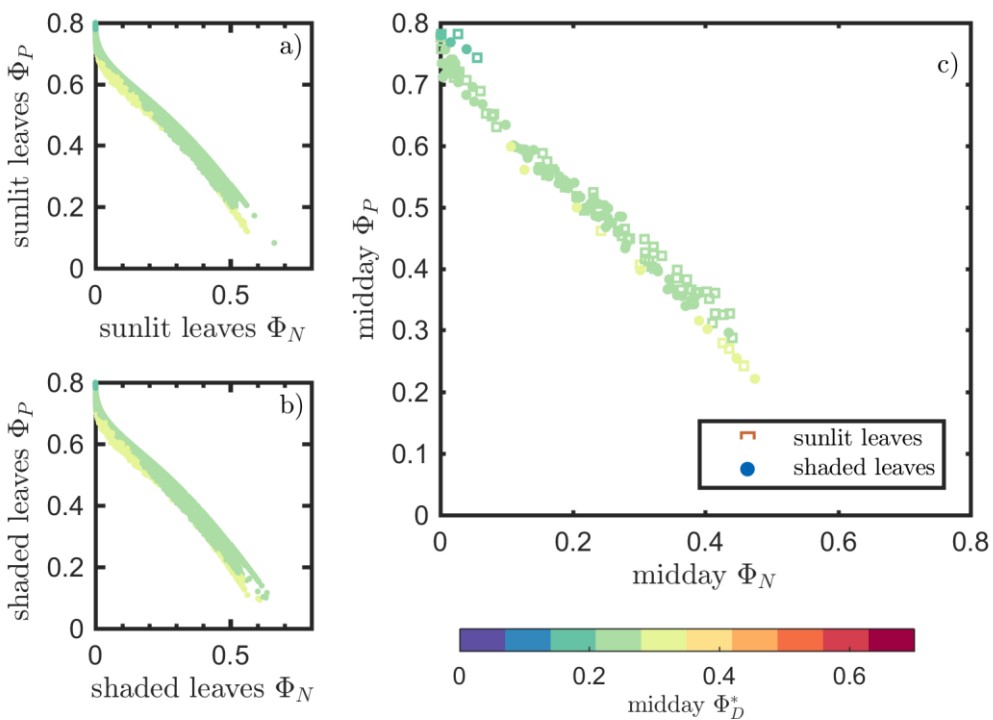

Figure 10: Reproduction of Fig. 7 with simulated variables from the biochemical model of Van der Tol et al. (2014).

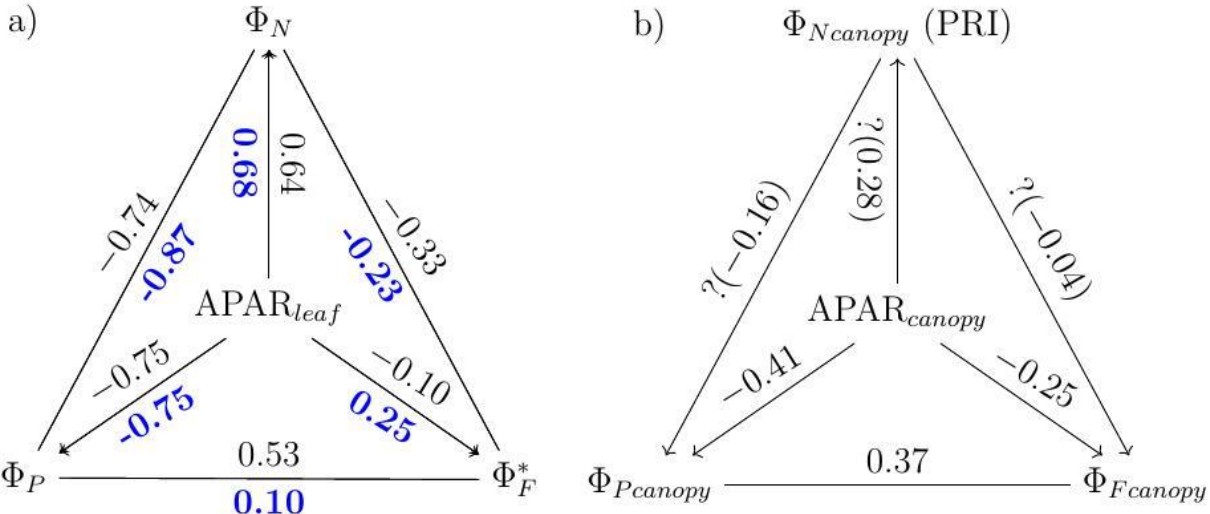

**Figure 11: Pearson correlation coefficients between absorbed PAR (APAR_leaf and APAR_canopy), and light use efficiencies for all data obtained for a corn canopy across the 2017 growing season at both leaf (a) and canopy levels (b). Light use efficiency of photochemistry ($\Phi_P$), relative fluorescence emission efficiency ($\Phi_F^*$), and efficiency of variable heat dissipation ($\Phi_N$) of sunlit leaves and shaded leaves (indicated in bold, blue text) during daytime (9:00 to 17:00) are obtained from *in situ* active fluorescence measurements made on individual leaves. Canopy light use efficiency of photochemistry ($\Phi_{Pcanopy}$) and of fluorescence ($\Phi_{Fcanopy}$) are approximated by GPP/ APAR_canopy and $F_{760}$/APAR_canopy respectively. PRI is used as an indicator of canopy light use efficiency of variable heat dissipation ($\Phi_{Ncanopy}$), but the exact values of $\Phi_{Ncanopy}$ are unknown (noted with "?" markers). The leaf-level and canopy-level variables had 10-minute and half-hour resolutions, respectively.**

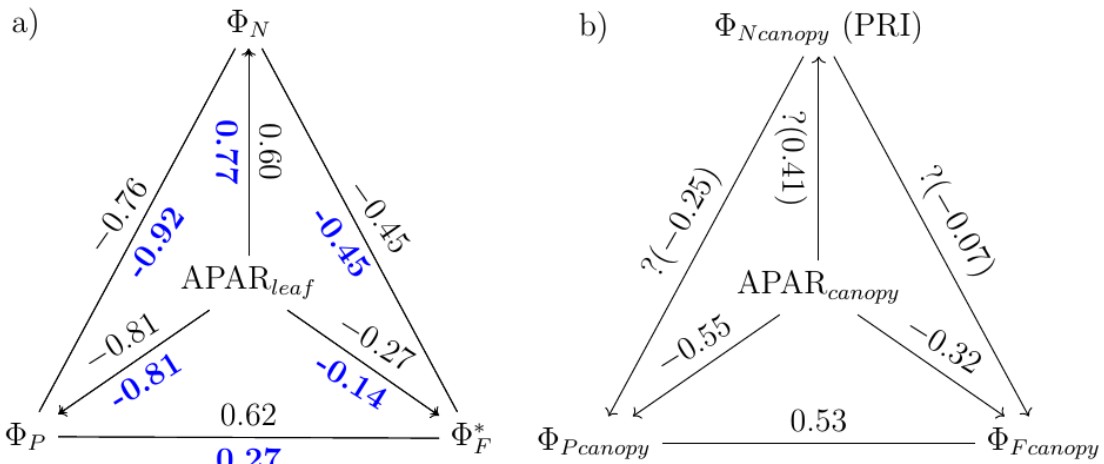

**Figure 12. Reproduction of Fig. 11 with only midday measurements (11:00-14:00). Data correspond to subsamples previously shown in Figs. 3e, 6c, and 7c.**

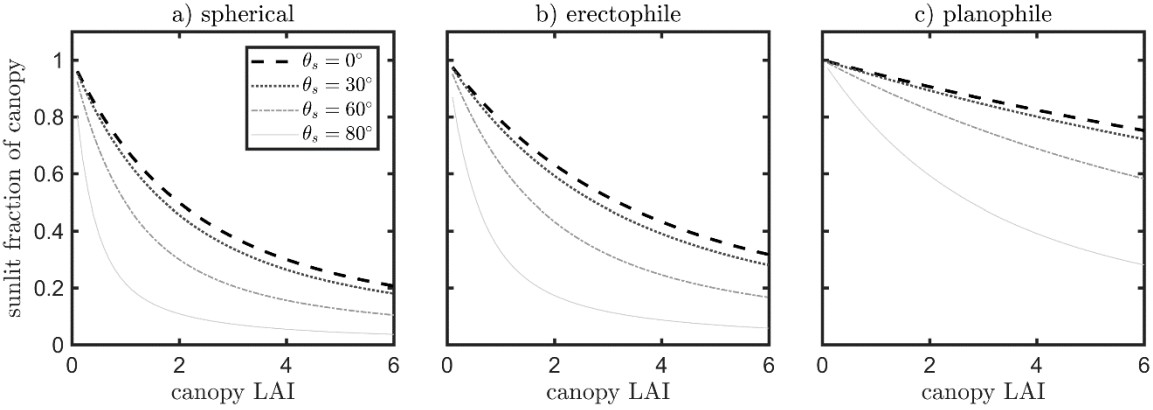

 **Figure 13: Fraction of sunlit canopy changing with canopy LAI and solar zenith angle ($\theta_s$) for canopy with spherical (a), erectophile (b) and planophile (c).**

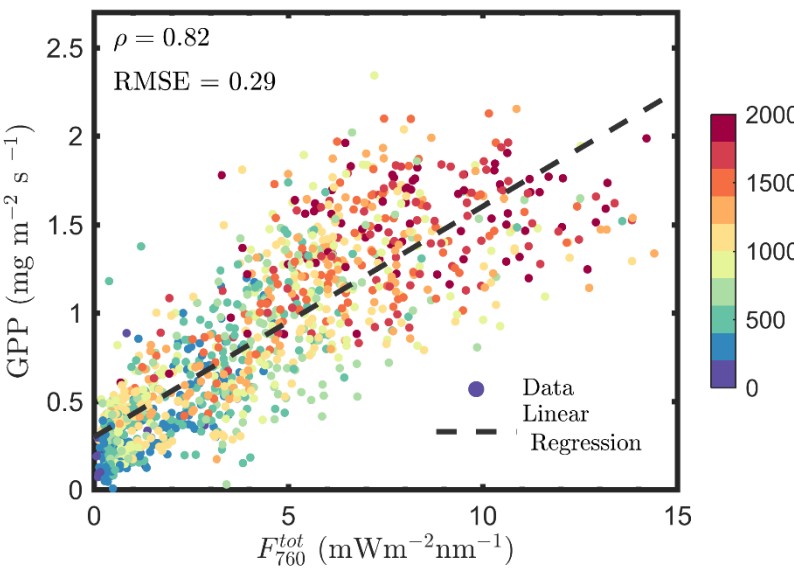

**Figure 14: Relationships between far-red total emitted SIF by the canopy ($F_{760}^{tot}$) and GPP. $F_{760}^{tot}$ was estimated by using the fluorescence correction vegetation index (FCVI).**

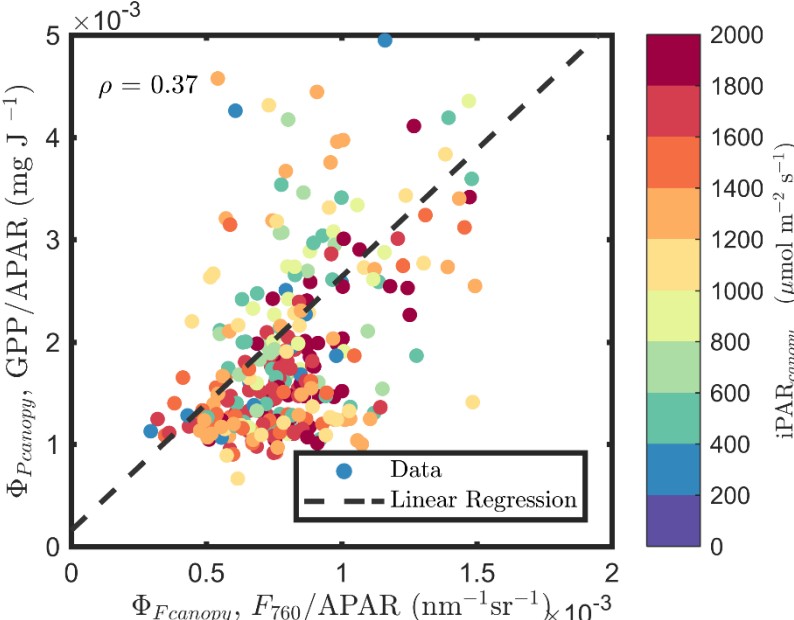

**Figure A1: Relationships between $\Phi_{Fcanopy}$ and $\Phi_{Pcanopy}$, estimated as $F_{760}$/APAR and GPP/APAR, respectively, of a corn canopy in the 2017 growing season with half-hour temporal resolution during daylight hours.**

1055

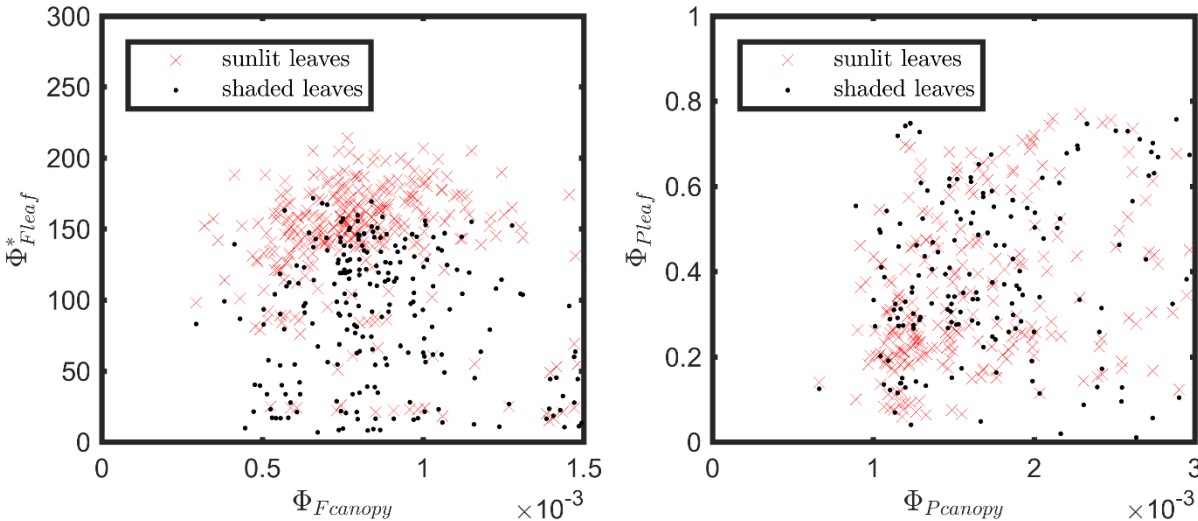

**Figure A2: Relationships between leaf and canopy $\Phi_F$ (a), and leaf and canopy $\Phi_P$ (b). $\Phi_{Fcanopy}$ and $\Phi_{Pcanopy}$ were estimated as $F_{760}$/APAR and GPP/APAR, respectively. $\Phi_{Fleaf}$ and $\Phi_{Pleaf}$ were derived from MoniPAM active fluorescence measurements.**