# Peer review of "Unravelling the physical and physiological basis for the solar-induced chlorophyll fluorescence and photosynthesis relationship using continuous leaf and canopy measurements of a corn crop"

_Biogeosciences, 2020_

## Referee Comment (RC1) · Anonymous Referee #1 · 29 Sep 2020

Review of "Unravelling the physical and physiological basis for the.." by P. Yang et al.

In this study Peiqi Yang and co-authors analyze observations from corn field during one growing season, where chlorophyll fluorescence (ChlF) has been measured both actively (MONIPAM) and passively (SIF) and relate the timeseries of these variables to gross primary production (GPP) measured from a flux tower. As the title says, they aim to study the relationship between GPP and the ChlF, a very important topic for a wider research community that now is using the novel SIF observations to estimate GPP. I find this manuscript very suitable for this journal and of interest to many.

The authors find out, that the correlation between GPP and SIF is small, once the

effect of absorbed PAR in canopy has been removed from the relationship. At the leaf level they found that the role of thermal dissipation was important factor influencing the relationship between fluorescence and photosynthesis yields. Also, they show the different functionalities of sunlit vs. shaded leaves in these respects and bring up the need to take them into account in the modelling efforts.

Major comments

I find the manuscript well-written and the figures clear. At times the text was a bit unspecific and challenging to understand (most of my comments are requirements for clarifications) and in the Discussion it was at times difficult to know, whether it was the leaf-level or canopy-level results being discussed. I'm sure the authors can overcome these issues with a bit more work on the text.

Of the things that were not discussed, I had few issues coming to my mind, that the authors might want address in the revised version. The title is very general, but the only plant being studied is corn, that is a C4 plant and might have a more linear relationship between SIF and GPP than C3 plants. Is this something worth mentioning somewhere and the possible differences related to C3 plants?

In the discussion of light environment, it is not mentioned that the tree canopies etc will have a more complex radiative transfer. Is the sunlit-shaded -separation something that is being recommended for crop canopies or is that something you consider sufficient also for more complex canopy structures?

You mention, that the correlation between photosynthetic and fluorescence yields estimated from canopy level had no correlation between the variables in the leaf level. The passive and active measurements anyhow differ in the very basics, i.e. in the passive you use just one wavelength, while MONIPAM gives you a spectrally integrated signal. Maybe you could also mention this?

Minor comments

l. 20: You mention, that the link between GPP and SIF is much weaker after taking into account iPAR and fAPAR. The correlation is below 0.30, so it is maybe even negligible. Maybe you could write this number here (because now it sounds, like there would still be definitely a functional link).

l. 20-22: Actually the positive correlation was present for sunlit leaves in well-illuminated conditions, whereas it was negative in the low-light conditions, was it? Maybe you could add that here.

l. 32: Eddy covariance measures the net flux, not GPP. This is not now obvious from the text.

l. 83: Sorry, what is 'fluorescence quenching'? And what maximum level are you referring to here? Maybe this sentence could be rephrased.

l. 116: Should it be 'carbon fluxes' instead of 'crop fluxes'?

l. 138: Not exactly clear, how the interpolation goes above the maximum observed value.

l. 144: Sometimes 'MoniPAM', few times 'MONIPAM'. The writing could be uniform throughout the text.

l. 161, section 2.4: Later you use PRI also, but you don't introduce its calculation.

l. 215: Maybe you could show the equation for photosynthetic LUE here. It is not necessarily clear to which variable you're referring to here, so that would help. This is unclear, because in line 223 you say you calculate variables using only leaf temperature and radiation intensity as input, but here you say that this variable is dependent on many different input variables.

l. 224: Would you have a reference for the crops Vcmo value? Which temperature response are you using for it?

Section 3.1. Are there changes in the LAI values during the growing season and is

there an increase in the senescent material in the field of view during the last development stage? The seasonal cycle of the observations is not shown. Therefore it is a bit difficult to judge, from which time period certain points in the e.g. Fig. 1 are.

Fig. 3: I miss having ticks in this figure. Especially, as the subplots don't have numbers. Adding ticks would help readability.

l. 278: "an order of magnitude improvement of 13%" - just wondering, if a higher correlation between SIF and GPP is necessarily 'an improvement', I'd tend to think it is just a higher correlation. Also, is increase of 13% 'an order of magnitude' size?

Fig. 4: A suggestion for this figure would be to make the panels bigger and include both sunlit and shaded in the same figure, shaded e.g. in dashed line. This would make comparison between the two easier (if it doesn't get far too busy plot).

l. 298: Is this midday dip of FiiN more occurring only in the sunlit leaves? Overall, when discussing Fig. 4 you don't mention differences between sunlit and shaded leaves. If they are similar in their dynamics, that's also maybe worth noting. There anyhow seems to be differences, that might be interesting, e.g. FiiPshade maybe goes lower fast during senescence, FiiFshade has lower values than FiiFsunlit, even though other components are perhaps on pretty similar levels.

l. 305-306: Maybe you can share some numbers about nighttime FiiF, as I find it difficult to see 'clear' decrease in these values. Is your sentence referring to this picture or the whole timeseries? For the sunlit leaves, it seems that during the young and mature stages there are nights with some higher values, but the overall level (at least as far as I can try to read the figure) is not that different. I'm not arguing your claim, but maybe you can back that up a bit.

l. 313: So, is the Fig. 5 for the sunlit or shaded leaves?

l. 314: 'evident... increased through the growing season': to me this sentence sounds that there is increase between all young – mature – senescence -stages. For the

nighttime, yes, there is a definite change during senescence compared to other stages. But the daytime values during senescence don't then seen lower, and then if there is a change in daytime values between young and mature stages is not so clear, as there is daily variation. For the Fig. 5 you chose 'representative' day for the pie chart. Could you tell on what conditions you chose this day? Did it have certain meteorological conditions or was it just similar as most other days?

I find it a bit annoying that you show days 193-197 in Fig. 4 and 192-196 in Fig. 5. It doesn't help in comparison. Was there a special reason you chose to show differing time periods?

l. 325-327: Are these numbers for the contributions correct? Based on numbers on the pie chart, I'd say different (but as mentioned below, I cannot read them clearly).

l. 341: Sorry, what does your 'seasonally averaged' means?

l. 345: A bit confusing, that you are here referring to subplots 6a and b, but the values are from the averaged plot 6c (and your point also).

l. 349: Should this be 6a (for FiiP and FiiF relationship)?

l. 350: You write in response to incoming light, but the color code here is for FiiN? If you want to emphasize 'to incoming light', maybe you can say something about that how it is related to this.

l. 351-353: Actually the arrow for the shaded leaves doesn't necessarily show the response to sustained heat dissipation so well, as the highest FiiD levels are not on the lowest levels (Fig. 7b). Also, yes, the responses between sunlit and shaded seem pretty similar, but just by looking, maybe the slopes (FiiP vs FiiN) in sunlit leaves change between the colored groups and not so in shaded leaves.

l. 363: Do you get the value 65% from the Fig. 7c? If so, you could clearly state which value you are referring to. (These larger variations in FiiP are also more present in lower FiiN values, logically. . .)

l. 376: Sorry, not clear what you mean by 'these trends'. The mentioned values were from half-hourly values and you mean that similar behavior is visible in seasonal and diurnal values?

l. 391: So, did you exclude measurement points from drought conditions from the dataset? Based on what conditions was that made? Or was the plot irrigated to start with, and you didn't have to worry about drought?

l. 394: So, did you find any ways that you could parameterize sustained heat dissipation, so that you could model it during the growing season?

l. 454: Your point here is that heat dissipation is more directly connected to photosynthetic lue than fluorescence emission to what.. heat dissipation? This sentence is a bit unclear, please rephrase. Are you here referring to Fig. 7a or 7b, are you talking about leaf or canopy level? Earlier you mentioned that give some doubt to PRI and show its correlations with a question mark. So this would be more about leaf level?

l. 456: 'photosynthetic energy', what do you actually mean by this? Is this the absorbed light energy in the photosynthetically active region?

l. 456: So are you now only referring to the study be Heber et al, or what did you see in your diurnal results? Or is the diurnal scale visible in 6c (but the relationship is not positive for shaded leaves)? To my understanding the review by Heber concentrates on mosses and lichens, quite different plants than corn. Maybe you could better clarify what is the meaning for you of this reference and how it related to your results?

l. 482: Sorry, what is the LUE-GPP relationship mentioned here?

l. 502: Often, when a model separates the canopy into sunlit and shaded fractions, it is called a two-leaf model (such a BEPS, e.g. Qiu et al 2019). Not 'two-big-leaf', even though to my understanding the idea is pretty much the same as you're here proposing.

l. 504: Sorry, is the a word missing in this sentence? Was shown what?
l. 502-509: In this paragraph you talk about LUE models and then mention SCOPE as an example of a more detailed model, but there are also large scale models of with varying degree of complexity (e.g. Parazoo et al, 2020), located between SCOPE and LUE models. Just mentioning, since now this paragraph offers maybe a quite narrow view.

l. 521: You mean that they (sunlit FiiF and FiiP) are more tightly connected than the FiiFshaded and FiiPshaded?

l. 560: So, you mean that the physiological traits of shade/light -adapted leaves would be good to be taken into account in SCOPE and other such models? It is not that clear how the above examination about the sunlit fraction depending on LAI and zenith angles really ties with the discussion. Could you maybe tie that better to the context?

Technical/typos

l. 68: 'improved the correlation', the correlation of SIF?

Table 1: Also add here how you measure the PRI.

Fig. 1: The a) and b) seem to be flipped.

l. 279: Why do you talk about 'mid-morning'? Your morning seems to end at midday, not to 'late morning'...

Fig. 4: Maybe change some y-axis labels for the right side for better readability?

Fig. 5: At least in my version the numbers in the pie charts are challenging to read. Could you improve the figure in that respect?

Fig. 6: Would you like to add a legend box? At first, the dot belonging to the legend might seem to be in the plot. Please, add ticks to subplots a and b.

l. 348: 'linear relationship'?

l. 355: Is 'expressed' the best word to use in this context?

Fig. 8c : The plotted symbols are below the subpanel name.

Fig. 10 caption: Do you mean in the second last sentence, that the values of FiiN-canopy are unknown or what?

Fig 10: Show the values with the same number of decimals, even if 0.1 is 0.10.

Fig. 11. In my copy it is not easy to differentiate the lines with zenith angle 30 and 0. At least in subpanel c) the legend also looks suspicious. If you want to have a w/b -figure here, could you maybe differentiate the lines with different widths or styles?

l. 444: So, are these now leaf level values?

References:

Parazoo, N. C., Magney, T., Norton, A., Raczka, B., Bacour, C., Maignan, F., Baker, I., Zhang, Y., Qiu, B., Shi, M., MacBean, N., Bowling, D. R., Burns, S. P., Blanken, P. D., Stutz, J., Grossmann, K., and Frankenberg, C.: Wide discrepancies in the magnitude and direction of modeled solar-induced chlorophyll fluorescence in response to light conditions, Biogeosciences, 17, 3733–3755, https://doi.org/10.5194/bg-17-3733-2020, 2020.

Qiu, B., Chen, J. M., Ju, W., Zhang, Q., and Zhang, Y.: Simulating emission and scattering of solar-induced chlorophyll fluorescence at far-red band in global vegetation with different canopy structures, Remote Sens. Environ., 233, 111373, https://doi.org/10.1016/j.rse.2019.111373, 2019.

---

## Referee Comment (RC2) · Anonymous Referee #2 · 26 Oct 2020

Yang et al. used a unique dataset comprising both active and passive measurements of fluorescence to explore the physical and physiological relationship between SIF and GPP. Considering the large amount work on the SIF-GPP relationships during the last few years, the work conducted is therefore relevant and it will be of interest for the scientific community working on remote sensing of GPP using SIF. The manuscript is a nice addition to the current body of literature and I think it is worth publishing. A few minor suggestions may be taken into account to improve the manuscript.

Line 167: Are the coefficients in Equation 2a obtained from Vina and Gitelson (2005)? If not, please add correct reference.

Lines 277-279 (Figure 3d-f): The data points are more disperse in the morning than in the afternoon. Please briefly discuss the possible reasons.

Figure 9: Shaded leaves exhibit higher light use efficiency of sustained heat dissipation than sunlit leaves, which is inconsistent with measured results (Figure 7). Briefly discuss the difference.

Line 464 (Figure 10b): The correlation coefficient between PRI and APARcanopy is 0.28, not -0.28 in Figure 10b, right?

In addition, many sentences can be improved, for example: Line 452: per leaf unit area → per unit leaf area

---

## Short Comment (SC1) · 27 Oct 2020

Review for "Unravelling the physical and physiological basis for the solar-induced chlorophyll fluorescence and photosynthesis relationship" by Yang et al.

General comments:

This manuscript used field measured leaf and canopy fluorescence and photosynthesis and investigated the physical and physiological basis of SIF-GPP relationship at a corn field. They found that APAR dominated the positive SIF-GPP relationship. They further used the continuous active fluorescence measurements from the MoniPAM system to analyze the relationship between fluorescence yield and photochemical yield at leaf scale and found a moderate correlation between the efficiencies of fluorescence emission and photochemistry for sunlit leaves but a weak correlation for shaded leaves.

The manuscript has some strength. The major strengths are: (1) The author combined leaf-scale active fluorescence measurements to fully investigate the physiological basis of the SIF-GPP relationship which is lacking in many studies. (2) The authors are on top of the most recent literatures in this topic. The references used are up to date, and the authors had a very thorough summary of the past literatures. The manuscript is also well-written.

However there are several unclear points which should be addressed:

(1) The reliability of relative efficiency of the sustained heat dissipation ($\Phi D*$) calculation. In L210, the author claims that "Because $Fm$ was measured during the night in the absence of both reversible heat dissipation and photochemistry, a change in $Fm$ must be caused by a change in the sustained heat dissipation". But during night, there are still $\Phi N$ and $\Phi F$ from Fig. 5. I am concerned about the reliability of $\Phi D*$ calculation since to my knowledge, this calculation hasn't been used in previous studies. The author should provide more literature to back up this method.

(2) The data availability across the whole growing season is not provided. In L154, the author mentioned that they excluded 29 days rainy and cloudy data, but the whole period of available canopy data is not provided. The author could provide a time series of the SIF, GPP, APAR data in the supplementary. Also, the availability of the active PAM measurements is also not explicitly provided.

(3) The author reported the overall correlation between $\Phi Pcanopy$ and $\Phi Fcanopy$. It would be good that they provide the scatter plot and compare this with the leaf scale relationship.

(4) L423 They found no clear relationships between $\Phi Pcanopy$ vs. $\Phi P$ or $\Phi Fcanopy$ vs. $\Phi F*$. This result needs more explanation, such as this poor correlation is for sunlit leaves or for shaded leaves or both and what causes this poor correlation. Of course, they are from different levels (leaf vs canopy) and canopy structure plays a role here. Although fesc calculation still has large uncertainty, there are several methods proposed to quantify this term (e.g., NIRv/fPAR). The author should try to correct fesc effect and get canopy total $\Phi Fcanopy$ and compare with leaf $\Phi F*$.

(5) L440. They found progressive increase of sustained heat dissipation ($\Phi D*$) during senescence. In contrast with no seasonal variation of $\Phi N$. Why there is no seasonal variation of $\Phi N$? What factor determined the seasonal variation of $\Phi N$.

(6) L455. The author mentioned that reversible heat dissipation is responsible for the positive relationship between $\Phi F$ and $\Phi P$ at diurnal scale, but there is no diurnal relationship between $\Phi F$ and

$\Phi P$ in the current manuscript. The author only provided the seasonal and seasonal+diurnal relationships.

(7) L520. The author claimed that a stronger relationship between SIF and GPP for dense canopies is expected since $\Phi F$ sunlit and $\Phi P$ sunlit are moderately correlated. I am not convinced that dense canopy means the fraction of sunlit leaves is larger. Also, the poor correlation between SIF and GPP at senescent stage is probably due to the less data points and more uncertainty of the SIF retrieval.

(8) L528. The author claimed that under cloudy conditions, SIF-GPP relationship becomes worse. But this is opposite to the previous study from Yang et al. (2018) in a rice paddy. They found similar relationship under sunny and cloudy conditions. Why will diffuse condition lead to a worse SIF-GPP relationship?

(9) Overall, I feel that the link between MoniPAM active fluorescence and canopy SIF is weak and the author analyzed these two datasets separately. Although they used to SCOPE but only to model the leaf scale relationship. It would be good if the author can use the leaf measurements to run SCOPE and get canopy SIF and GPP and compare with observations.

Finally, I want to provide encouragements for this work. The general goal that this work aims to achieve is worth praising. I enjoyed the reading of this manuscript and it clearly shows the authors have been putting lots of efforts into the literature review. I can see that this work could have a good impact and contribution to this field if all the above concerns can be properly addressed. Thus I fully encourage moderate revision of this work. Meanwhile, please understand that a rigorous scrutiny is necessary here as this topic that you are addressing is very important and your conclusion can have a large impact for the general public's understanding about SIF and photosynthesis.

---

## Author Comment (AC2) · 2 Nov 2020

**Unravelling the physical and physiological basis for the solar-induced chlorophyll fluorescence and photosynthesis relationship using continuous leaf and canopy measurements of a corn crop**

**Response to reviewers' comments**

*Peiqi Yang, Christiaan van der Tol, Petya K. E. Campbell, Elizabeth M. Middleton*

*October, 2020*

**Anonymous Referee #2**

Yang et al. used a unique dataset comprising both active and passive measurements of fluorescence to explore the physical and physiological relationship between SIF and GPP. Considering the large amount work on the SIF-GPP relationships during the last few years, the work conducted is therefore relevant and it will be of interest for the scientific community working on remote sensing of GPP using SIF. The manuscript is a nice addition to the current body of literature and I think it is worth publishing. A few minor suggestions may be taken into account to improve the manuscript.

Response: We thank the reviewer for the encouraging feedback. We have revised the manuscript according to both reviewers' comments.

Line 167: Are the coefficients in Equation 2a obtained from Vina and Gitelson (2005)? If not, please add correct reference.

Response: The coefficients are not explicitly given in Vina and Gitelson (2005), but they can be obtained from the linear regression line on Fig. 3C in Vina and Gitelson (2005). We obtained these coefficients from Vina and Gitelson (2005) and also from Miao et al., (2018), in which the exact coefficients are given. The difference of the two sets of coefficients is very small (1.373 RededgeNDVI-0.172 vs. 1.37 RededgeNDVI-0.17). In the revised version, we have added this reference Miao et al., (2018) apart from Vina and Gitelson (2005).

Lines 277-279 (Figure 3d-f): The data points are more disperse in the morning than in the afternoon. Please briefly discuss the possible reasons.

Response: More disperse data points suggest lower correlation between SIF and GPP in the morning. We have discussed one possible reason in section 4.3, which is the fraction of diffuse light. However, we also believe that there are other possible reasons. For example, LUEp is generally more variable in the morning due to the quicker change of environmental conditions, such as air temperature and humanity, whereas LUEf may not change correspondingly. The different response of LUEp and LUEf to the rapidly changing environment may have caused the lower correlation.

Figure 9: Shaded leaves exhibit higher light use efficiency of sustained heat dissipation than sunlit leaves, which is inconsistent with measured results (Figure 7). Briefly discuss the difference.

Response: Thank you for this comment. The model does not simulate a different response to environmental variables between sunlit and shaded leaves (it considers just one type of leaf). The very small difference in simulated heat dissipation is because the sunlit leaves have a higher temperature than the shaded leaves. This affects the sustained heat dissipation in the model. We made a mistake in using the color map and have corrected it.

Line 464 (Figure 10b): The correlation coefficient between PRI and APARcanopy is 0.28, not -0.28 in Figure 10b, right?

Response: The correct value is 0.28. We have revised Fig. 10b in the revision.

In addition, many sentences can be improved, for example: Line 452: per leaf unit area → per unit leaf area

Response: We have checked and improved the overall quality of our writing.

---

## Author Response (AR1)

Unravelling the physical and physiological basis for the solar-induced chlorophyll fluorescence and photosynthesis relationship using continuous leaf and canopy measurements of a corn crop

**Response to editor's and reviewers' comments**

*Peiqi Yang, Christiaan van der Tol, Petya K. E. Campbell, Elizabeth M. Middleton*

*October, 2020*

**Content**

**Contents**

**Comments from the editor to the Author and our response**

Dear Dr. Yang and co-authors,

As you have seen by now, your work was assessed by three reviews. All three of the reviewers are generally encouraging of this work and note that it is potentially an important contribution to the literature on the SIF-GPP relationship. However, the number of requested clarifications and comments is extensive, so I recommend revising with major revisions.

Dear Dr. Konings, We thank you for your valuable time serving as the editor of our manuscript, and giving us the opportunity to improve our study.

Based on your author responses submitted so far, here are a few things to keep in mind as you revise the manuscript:

1) In your author responses, you have addressed a number of the requests for clarification by the three reviewers only by providing clarifying statements in the review document - in most cases, these requests for clarification should also lead to changes in the text to make sure the text is made more clear for the readers.

Response: We acknowledge the shortcomings of our response letter posted in the discussion forum. We had revised the text in the manuscript as we stated in the response letter, but did not know if it was appropriate to attach a track-change version of our revised manuscript. As suggested, we understand that the better way is to specify the changes in the response letter to make the revision more clearly for the readers, and we have modified our response letter by adding the specific changes.

2) When you send a revised manuscript with response to reviewer documents, please do not just say that you have clarified something in the text as you do in the responses you have pointed to so far, but include the actual new text in your response to reviewers document. Otherwise it is quite difficult to determine that you have actually satisfactorily addressed the comments.

Response: Yes, we agree with your suggestions. We have added the specific changes to our response in the letter instead of just mentioning it. Furthermore, we have included the revised manuscript with the changes highlighted in the response letter.

3) Additionally, please make sure the abstract is consistent with the most important points in the text. For example, you plan to show the diurnal relationship between phi F and phi P only in the supplement, but this is still highlighted as a key finding in your abstract.

Response: Thanks for point this out. We have decided to include the diurnal correlations as the main results since they support our main findings of the study.

4) Lastly, please do pay attention to Prof. Guan's last comment. While there is nothing wrong with arguing that a reviewer suggestion is outside the scope of your manuscript, the underlying comment that the leaf-level and canopy-level results are not sufficiently well-integrated with each other in your analysis should still be addressed in your response, not just the suggestion to run SCOPE. Several other comments by Dr. Guan (#'s 5 and 8, for example) are also only partially addressed in the current author responses.

Response: We acknowledge that more comprehensive responses are needed to better address several comments raised by Dr. Guan in the response letter, although we did discuss them in the revised manuscript. As we mentioned in our response to your comment #1 and #2, we have incorporated more detailed reply to all the reviewers' comments.

Please keep the above comments in mind as you prepare your revised manuscript and a final response document. I look forward to receiving the revisions.

Best regards, Alex Konings

**Comments from Anonymous Referee #1 and our response**
**General comments**

In this study Peiqi Yang and co-authors analyze observations from corn field during one growing season, where chlorophyll fluorescence (ChlF) has been measured both actively (MONIPAM) and passively (SIF) and relate the timeseries of these variables to gross primary production (GPP) measured from a flux tower. As the title says, they aim to study the relationship between GPP and the ChlF, a very important topic for a wider research community that now is using the novel SIF observations to estimate GPP. I find this manuscript very suitable for this journal and of interest to many.

The authors find out, that the correlation between GPP and SIF is small, once the effect of absorbed PAR in canopy has been removed from the relationship. At the leaf level they found that the role of thermal dissipation was important factor influencing the relationship between fluorescence and photosynthesis yields. Also, they show the different functionalities of sunlit vs. shaded leaves in these respects and bring up the need to take them into account in the modelling efforts.

Response: We thank the reviewer for the positive and encouraging feedback. The reviewer's comments and suggestions are constructive and have helped us to improve the manuscript substantially.

**Major comments**

1.  I find the manuscript well-written and the figures clear. At times the text was a bit unspecific and challenging to understand (most of my comments are requirements for clarifications) and in the

Discussion it was at times difficult to know, whether it was the leaf-level or canopy-level results being discussed. I'm sure the authors can overcome these issues with a bit more work on the text.

Response: We have improved the clarity throughout the manuscript with the help of the reviewers' specific comments. Most of figures have been revised for a better readability, e.g., by adding more ticks, rearranging subpanels in a figure and adjusting size of the figures.

2. Of the things that were not discussed, I had few issues coming to my mind, that the authors might want address in the revised version. The title is very general, but the only plant being studied is corn, that is a C4 plant and might have a more linear relationship between SIF and GPP than C3 plants. Is this something worth mentioning somewhere and the possible differences related to C3 plants?

Response: Yes, it is a valid point. We only examined a corn crop, which is a C4 species. In the revised version, we have first specified that a corn crop is studied in the title and mentioned the main characteristic of corn in the introduction, and then included a short discussion on the difference between C3 and C4 crops:

*"The investigated crop has a C4 photosynthetic pathway, in which dark and light reactions are separated, and the carboxylation takes place under a high $CO_2$ concentration. This strongly suppresses photorespiration in C4 vegetation, resulting in a higher water use efficiency and lower sensitivity to heat and higher vapour pressure deficit than C3 vegetation. Liu et al. (2017) reported that the GPP–SIF relationship was much higher for C4 crops. They showed that $\Phi_{Fcanopy}$ of the C3 and C4 crops were similar but $\Phi_{Pcanopy}$ of C4 corn was much higher than C3 wheat. Because of a different photosynthetic pathway and the contribution of photorespiration, the SIF-GPP relationship of C3 vegetation is more complicated in the corn crop examined in this study".*

3. In the discussion of light environment, it is not mentioned that the tree canopies etc will have a more complex radiative transfer. Is the sunlit-shaded -separation something that is being recommended for crop canopies or is that something you consider sufficient also for more complex canopy structures?

Response: We believe that the separation of sunlit and shaded leaves is needed for complex canopies as well. The approach for separation we used is based on a turbid medium assumption, and can be directly applied to structurally simple canopies, such as corn. However, in the revised version, we have acknowledged that for structurally complex canopies, our approach can only serve as a first-order estimation, and additional structural characteristics should be included in separating sunlit and shaded leaves. One of the most important characteristics is the clumping index, because the clumping of leaves affects the gap probability in the vegetation canopy, the light penetration, and thus the sunlit fraction of the vegetation canopy.

4. You mention, that the correlation between photosynthetic and fluorescence yields estimated from canopy level had no correlation between the variables in the leaf level. The passive and active measurements anyhow differ in the very basics, i.e. in the passive you use just one wavelength, while MONIPAM gives you a spectrally integrated signal. Maybe you could also mention this?

Response: The reviewer is absolutely right that the passive SIF is measured at one narrow band while the active fluorescence is an integral over a wide band. Although we believe that the difference between leaf and canopy measurements is mainly due to the canopy structure effects, we agree it is worth mentioning about the difference between MoniPAM and passive fluorescence measurements. Hence, we have discussed the difference as suggested in the revised version:

*"It is worth noting that active fluorescence measurements are spectrally integrated signals, whereas canopy passive SIF observations are obtained at one wavelength. As a result, the leaf-level*

*fluorescence emission and photosynthetic light use efficiencies derived from active fluorescence measurements differ spectrally from the canopy-level efficiencies (ΦFcanopy and ΦPcanopy). This difference may also play a role in upscaling leaf-level to canopy-level relationship between ΦF and ΦP."*

**Minor comments**

5. l. 20: You mention, that the link between GPP and SIF is much weaker after taking into account iPAR and fAPAR. The correlation is below 0.30, so it is maybe even negligible. Maybe you could write this number here (because now it sounds, like there would still be definitely a functional link).

Response: Agreed. We have revised it accordingly.

*"the remaining correlation between far-red SIF and GPP due solely to the functional link between fluorescence and photosynthesis at the photochemical level was much weaker (ρ = 0.30)."*

6. l. 20-22: Actually the positive correlation was present for sunlit leaves in well illuminated conditions, whereas it was negative in the low-light conditions, was it? Maybe you could add that here.

Response: Agreed. We have revised it accordingly.

*"Active leaf-level fluorescence measurements revealed a moderate positive correlation between the efficiencies of fluorescence emission and photochemistry for sunlit leaves in well-illuminated conditions but a weak negative correlation in the low-light condition, and which was negligible for shaded leaves."*

7. l. 32: Eddy covariance measures the net flux, not GPP. This is not now obvious from the text.

Response: We have specified that eddy covariance flux towers provide point measurements of net carbon flux in the revision.

8. l. 83: Sorry, what is 'fluorescence quenching'? And what maximum level are you referring to here? Maybe this sentence could be rephrased.

Response: Fluorescence quenching refers to any process that decreases the fluorescence of a sample. The maximum level refers to the status when the photochemical pathway is completely inhibited (e.g. by using a saturating light). We have rephrased the sentence in the revised version.

"The relationship between the photochemical-level photosynthetic light use efficiency (ΦP) and fluorescence reduction (i.e., quenching) was described with the Genty equation as (Fm- Fs)/Fm (Genty et al., 1989) which compares the relative fluorescence change from a steady state (Fs) to its maximal level (Fm) when the photochemical pathway is completely inhibited (e.g., by using a saturating light)."

9. l. 116: Should it be 'carbon fluxes' instead of 'crop fluxes'?

Response: Yes. We have corrected the mistake.

10. l. 138: Not exactly clear, how the interpolation goes above the maximum observed value.

Response: We used both extrapolation and interpolation. It has been clarified in the revision.

11. l. 144: Sometimes 'MoniPAM', few times 'MONIPAM'. The writing could be uniform throughout the text.

Response: We have revised accordingly by consistently using MoniPAM.

12. l. 161, section 2.4: Later you use PRI also, but you don't introduce its calculation.

Response: We introduced its calculation in section 3.4, but we agree that it is better to mention the calculation of PRI in the method section. Hence, we have added a sentence about its calculation in section 2.4.

13. l. 215: Maybe you could show the equation for photosynthetic LUE here. It is not necessarily clear to which variable you're referring to here, so that would help. This is unclear, because in line 223 you say you calculate variables using only leaf temperature and radiation intensity as input, but here you say that this variable is dependent on many different input variables.

Response: We agree this requires further clarification. We meant that all the input variables were required, but we have field measurements of the two most important variables (leaf temperature and radiation). For the remaining variables, the model default values were used. The photosynthesis model (the FvCB model) is relatively simple, but still requires some efforts to explain. Instead of providing a set of equations of the photosynthesis model, we have rephrased the text on the model simulation to make our simulation settings clear.

*"The two most influential model input variables, leaf temperature and incoming radiation, were measured by using the field measurements MoniPAM. Vcmo was set to 30 μmol m-2 s-1 at 25 °C, a recommended value for the corn crop (Houborg et al., 2013; Wullschleger, 1993; Zhang et al., 2014)."*

14. l. 224: Would you have a reference for the crops Vcmo value? Which temperature response are you using for it?

Response: The value of Vcmo varies with temperature. In Zhang et al (2014), the estimated Vcmo values of corn range from 11 to 75 with an average of 37 u mol m-2 s-1. In Houborg et al. (2013), the reported Vcmax at 25 °C (i.e., Vcmo) of 11 to 48 umol m-2 s-1 for corn during the growing season.

Vcmo refers to the Vcmax at 25°C. For the temperature response of Vcmax, we use Collatz et al. (1992), namely Vcmax = Vcmo x 2.1^ ((temperature-25)/10). Additional constraints for extreme low and high temperature condition are considered but not shown in the equation, which can be found in Collatz et al. (1992).

15. Section 3.1. Are there changes in the LAI values during the growing season and is there an increase in the senescent material in the field of view during the last development stage? The seasonal cycle of the observations is not shown. Therefore it is a bit difficult to judge, from which time period certain points in the e.g. Fig. 1 are.

Response: Yes, there are changes in LAI and as well as increase of the senescent material during the last developmental stage. We have retrieved the values of LAI and senescent material from TOC reflectance by inverting a radiative transfer model. The results, however, are not directly related to the topic of the present manuscript. Therefore, we only show the retrieved values in the response letter to address the reviewer's comment. Please find them in the figure below.

As for the seasonal variation of observations, we have provided all the measurements of GPP, SIF, and MoniPAM measurements in a supplement. The link to the data is on the same page with the manuscript below the manuscript pdf icon (https://bg.copernicus.org/preprints/bg-2020-323/bg-2020-323-supplement.zip).

[Figure]

Fig. retrieved values of LAI and relative senescent material from TOC reflectance by inverting a radiative transfer model.

16. Fig. 3: I miss having ticks in this figure. Especially, as the subplots don't have numbers. Adding ticks would help readability.

Response: We have added the ticks to the figure as suggested.

17. l. 278: "an order of magnitude improvement of 13%" - just wondering, if a higher correlation between SIF and GPP is necessarily 'an improvement', I'd tend to think it is just a higher correlation. Also, is increase of 13% 'an order of magnitude' size?

Response: We agree with the reviewer's interpretation of the results and with the comment on the magnitude, and have revised it by removing the statement "an order of magnitude improvement of 13%".

*""As for the different times of a day, we found that their correlations were the strongest in the afternoon (ρ = 0.89) while ρ was only 0.76 when the data were acquired in the morning (Figs 3d vs. 3f).*

18. Fig. 4: A suggestion for this figure would be to make the panels bigger and include both sunlit and shaded in the same figure, shaded e.g. in dashed line. This would make comparison between the two easier (if it doesn't get far too busy plot).

Response: We have enlarged and merged the two panels in the revised revision.

19. l. 298: Is this midday dip of FiiN more occurring only in the sunlit leaves? Overall, when discussing Fig. 4 you don't mention differences between sunlit and shaded leaves. If they are similar in their dynamics, that's also maybe worth noting. There anyhow seems to be differences, that might be interesting, e.g. FiiPshade maybe goes lower fast during senescence, FiiFshade has lower values than FiiFsunlit, even though other components are perhaps on pretty similar levels.

Response: We found that this midday dip occurred in both sunlit and shaded leaves. This is a nice suggestion. We have added a paragraph about the comparison of sunlit and shaded leaves in section 3.2.

*"Although the sunlit and shaded leaves had similar seasonal and diurnal patterns, some interesting differences are observed. As expected, the radiation levels were higher for the sunlit leaves than for the shaded leaves, which produced higher ΦF\* for the sunlit leaves and slightly lower ΦP at the young and mature stages. In comparison to the difference in ΦF\*, the difference in ΦP was less pronounced. At the senescent stage ΦP of the shaded leaves was substantially lower than sunlit leaves despite receiving lower radiation, which normally would lead to higher ΦP. This could be attributed*

*to the different leaf ages and functionality of sunlit and shaded leaves; for example, shaded corn leaves senesce earlier than sunlit leaves. Additionally, ΦD\* of sunlit leaves was higher than the shaded leaves while ΦN of the sunlit and shaded leaves was similar."*

20. l. 305-306: Maybe you can share some numbers about nighttime FiiF, as I find it difficult to see 'clear' decrease in these values. Is your sentence referring to this picture or the whole timeseries? For the sunlit leaves, it seems that during the young and mature stages there are nights with some higher values, but the overall level (at least as far as I can try to read the figure) is not that different. I'm not arguing your claim, but maybe you can back that up a bit.

Response: We have included some numbers of PhiF to justify the reduction of nighttime PhiF. At the young stage, its value was around 60, while in the senescent stage it was 50.

21. l. 313: So, is the Fig. 5 for the sunlit or shaded leaves?

Response: This figure is for sunlit leaves. We have clarified in the caption of the figure.

22. l. 314: 'evident: : : increased through the growing season': to me this sentence sounds that there is increase between all young – mature – senescence -stages. For the nighttime, yes, there is a definite change during senescence compared to other stages. But the daytime values during senescence don't then seen lower, and then if there is a change in daytime values between young and mature stages is not so clear, as there is daily variation. For the Fig. 5 you chose 'representative' day for the pie chart. Could you tell on what conditions you chose this day? Did it have certain meteorological conditions or was it just similar as most other days?

Response: The reviewer is right. The diurnal variation should have been considered. Nevertheless, we think the argument is valid for daytime as well, because the daytime and nighttime sustained heat dissipations are the same, and an increase of PhiD is observed for both daytime and nighttime, which leads to a decrease of PhiP in a seasonal cycle. That is to say, because of the increase of PhiD from young and mature to senescent stage, it is expected that both nighttime and daytime PhiP decreases despite of its diurnal variation. The three 'representative' days were selected on the condition that clouds effects are negligible according to the iPAR measurements, so they were representative for clear sky conditions

23. I find it a bit annoying that you show days 193-197 in Fig. 4 and 192-196 in Fig. 5. It doesn't help in comparison. Was there a special reason you chose to show differing time periods?

Response: Sorry for our mistake. They should have been the same. We have revised Fig. 5 to make them consistent.

24. l. 325-327: Are these numbers for the contributions correct? Based on numbers on the pie chart, I'd say different (but as mentioned below, I cannot read them clearly).

Response: Sorry for the slight inconsistency between the figure and the text. We have revised the numbers in the text accordance with the values in the figure. The differences were small (less than 2%) and did not affect our arguments in the text.

25. l. 341: Sorry, what does your 'seasonally averaged' means?

Response: 'Seasonally averaged' is reductant and misleading here. We have deleted it.

26. l. 345: A bit confusing, that you are here referring to subplots 6a and b, but the values are from the averaged plot 6c (and your point also).

Response: The reviewer is right and it should be Fig. 6c. We have revised the text accordingly.

27. l. 349: Should this be 6a (for FiiP and FiiF relationship)?

Response: Yes. We have revised the text accordingly.

28. l. 350: You write in response to incoming light, but the color code here is for FiiN? If you want to emphasize 'to incoming light', maybe you can say something about that how it is related to this.

Response: As the reviewer suggested, we have clarified that PhiN increased with increasing incoming PAR as shown in Fig. 4.

29. l. 351-353: Actually the arrow for the shaded leaves doesn't necessarily show the response to sustained heat dissipation so well, as the highest FiiD levels are not on the lowest levels (Fig. 7b). Also, yes, the responses between sunlit and shaded seem pretty similar, but just by looking, maybe the slopes (FiiP vs FiiN) in sunlit leaves change between the colored groups and not so in shaded leaves.

Response: The reviewer is correct on the difference between sunlit and shaded leaves. We have acknowledged this difference in the revised version. Please see our response to your comment #19.

30. l. 363: Do you get the value 65% from the Fig. 7c? If so, you could clearly state which value you are referring to. (These larger variations in FiiP are also more present in lower FiiN values, logically: : :)

Response: Yes, the values are from Fig. 7c. PhiP can vary from 0.37 to 0.61 when PhiN was around 0.05. We have included this additional information in the revised version to improve the clarity.

31. l. 376: Sorry, not clear what you mean by 'these trends'. The mentioned values were from half-hourly values and you mean that similar behavior is visible in seasonal and diurnal values?

Response: We were referring to the observed reduction of the correlation between PhiF and PhiP. We have revised this sentence as

*''The reduction of the correlation between PhiP and PhiF was caused by diurnal variations in PhiN as well as seasonal variations in both PhiN and PhiD.''*

32. l. 391: So, did you exclude measurement points from drought conditions from the dataset? Based on what conditions was that made? Or was the plot irrigated to start with, and you didn't have to worry about drought?

Response: Sorry for the confusion. We meant that the drought effects were not included in the simulation but were very likely present in the field measurements. We have clarified this in the revision.

*"Furthermore, we did not include changes in leaf display geometry induced by low water stress (i.e., drought) in the simulations, but it is a common phenomenon in corn plants in the field."*

33. l. 394: So, did you find any ways that you could parameterize sustained heat dissipation, so that you could model it during the growing season?

Response: At this point, we have not found a convincing way to parameterize the sustained heat dissipation, because its controlling factors are still not clear. Our study only shows the seasonal variation, which is related to the change of pigment pools, but we don't have a certain answer on this.

34. l. 454: Your point here is that heat dissipation is more directly connected to photosynthetic lue than fluorescence emission to what.. heat dissipation? This sentence is a bit unclear, please rephrase. Are you here referring to Fig. 7a or 7b, are you talking about leaf or canopy level?

Earlier you mentioned that give some doubt to PRI and show its correlations with a question mark. So this would be more about leaf level?

Response: Yes, it is about leaf level. We have rephrased this part by stating the role of sustained and reversible heat dissipation on the diurnal and seasonal variation of leaf photosynthetic light use efficiency, respectively.

*"Compared to the relationship between leaf fluorescence emission efficiency, total heat dissipation (both D and N) provided a robust and direct indicator of leaf photosynthetic light use efficiency (Fig. 7). In particular, the variation of reversible heat dissipation better explains the diurnal variation of leaf photosynthetic light use efficiency, whereas the sustained heat dissipation contributes to the seasonal variation."*

35. l. 456: 'photosynthetic energy', what do you actually mean by this? Is this the absorbed light energy in the photosynthetically active region?

Response: Yes, we have changed it to 'absorbed photosynthetically active radiation'.

36. l. 456: So are you now only referring to the study be Heber et al, or what did you see in your diurnal results? Or is the diurnal scale visible in 6c (but the relationship is not positive for shaded leaves)? To my understanding the review by Heber concentrates on mosses and lichens, quite different plants than corn. Maybe you could better clarify what is the meaning for you of this reference and how it related to your results?

Response: We have added several more relevant papers showing the dominant role of reversible heat dissipation in various vegetation. Our study confirms this with field measurements and finds PhiN is responsible for regulating the correlation between PhiP and PhiF. The reviewer is right that the positive relationship exists for sunlit leaves. We have addressed this in the revision.

*"Reversible heat dissipation is the main regulating mechanism for the dissipation of absorbed photosynthetically active radiation (Adams et al., 1989; Demmig-Adams et al., 1996; Heber et al., 2006; Huang et al., 2006). Our study confirms its dominant role for the corn crop with field measurements and finds that the reversible heat dissipation is responsible for the positive relationship between $\Phi_F$ and $\Phi_P$ of sunlit leaves at diurnal scales, though less so at seasonal scales when sustained heat dissipation is dominant (Fig. 6)"*

37. l. 482: Sorry, what is the LUE-GPP relationship mentioned here?

Response: It should be the relationship of photosynthetic light use efficiencies at both leaf and canopy levels.

38. l. 502: Often, when a model separates the canopy into sunlit and shaded fractions, it is called a two-leaf model (such a BEPS, e.g. Qiu et al 2019). Not 'two-big-leaf', even though to my understanding the idea is pretty much the same as you're here proposing.

Response: We are aware that both 'two-leaf' and 'two-big-leaf' models are used interchangeably (Dai et al, 2004; Luo et al., 2018; Parazoo et al, 2020), and agree with the reviewer's suggestion. In the revised version, we have used 'two-leaf' models and noted that 'two-big-leaf' was also used in literature.

39. l. 504: Sorry, is the a word missing in this sentence? Was shown what?

Response: Yes, the missing part is 'an improved correlation with LUE'.

40. l. 502-509: In this paragraph you talk about LUE models and then mention SCOPE as an example of a more detailed model, but there are also large scale models of with varying degree of

complexity (e.g. Parazoo et al, 2020), located between SCOPE and LUE models. Just mentioning, since now this paragraph offers maybe a quite narrow view.

Response: Thanks for the nice review article. We have incorporated a more compressive discussion of the existing models, such as SCOPE, BEPS-SIF, BETHY-SCOPE and DART.

*"Qiu et al, (2019) incorporated a fluorescence simulation in the Boreal Ecosystem Productivity Simulator (BEPS, Liu et al., 1997), which is a two-leaf process-based model. More classes of leaves with varying ambient temperatures and incident radiation levels can be examined using more explicit models, such as SCOPE (Soil-Canopy-Observation of Photosynthesis and Energy fluxes, Van Der Tol et al., 2009), BETHY-SCOPE (the Biosphere Energy Transfer Hydrology model coupled with SCOPE, Norton et al., 2018) or DART (the Discrete Anisotropic Radiative Transfer model, Gastellu-Etchegorry et al., 2017)."*

41. l. 521: You mean that they (sunlit FiiF and FiiP) are more tightly connected than the FiiFshaded and FiiPshaded?

Response: Yes, we have clarified as suggested.

42. l. 560: So, you mean that the physiological traits of shade/light -adapted leaves would be good to be taken into account in SCOPE and other such models? It is not that clear how the above examination about the sunlit fraction depending on LAI and zenith angles really ties with the discussion. Could you maybe tie that better to the context?

Response: Yes, the examination of the sunlit fraction changing with canopy structure and zenith angles provides a prediction at a single moment. To account for the different physiological traits of shade/light -adapted leaves, we could predict the light distribution inside the canopy with varying sun positions (e.g., a diurnal cycle). In this way, sun-adapted and shade-adapted leaves can be differentiated according to the probability of being illuminated for a longer period instead of assuming a steady state. A leaf is considered as sun-adapted when it is almost always illuminated at various sun positions or different time in a day. Furthermore, different physiological traits of sun-adapted and shade-adapted can be taken into account in the model.

**Technical/typos**

43. l. 68: 'improved the correlation', the correlation of SIF?

Response: Yes. It has been clarified in the revision.

*"improved the correlation between SIF and APAR but not GPP"*

44. Table 1: Also add here how you measure the PRI.

Response: We have added the calculation of PRI in section 2.4 in the revision.

45. Fig. 1: The a) and b) seem to be flipped.

Response: The reviewer is right. We have swapped their positions in the revision.

46. l. 279: Why do you talk about 'mid-morning'? Your morning seems to end at midday, not to 'late morning': : :

Response: We have changed mid-morning and mid-afternoon to morning and afternoon.

47. Fig. 4: Maybe change some y-axis labels for the right side for better readability?

Response: We have moved all the y-axis labels to the right side to improve the readability.

48. Fig. 5: At least in my version the numbers in the pie charts are challenging to read. Could you improve the figure in that respect?

Response: Yes, we have changed the figure from 1.5 column width to double column (full width). Additionally, we have increased the resolution of the figure.

49. Fig. 6: Would you like to add a legend box? At first, the dot belonging to the legend might seem to be in the plot. Please, add ticks to subplots a and b.

Response: Yes, it is a nice suggestion. We have added a legend box and ticks to the figure in the revised version.

50. l. 348: 'linear relationship'?

Response: Agreed and revised accordingly.

51. l. 355: Is 'expressed' the best word to use in this context?

Response: We have used 'fully manifest' for a clearer meaning.

52. Fig. 8c : The plotted symbols are below the subpanel name.

Response: We have moved the subpanel names to the other side to avoid overlapping with the symbols.

53. Fig. 10 caption: Do you mean in the second last sentence, that the values of FiiNcanopy are unknown or what?

Response: Yes, there was a typo in FiiFcanopy and it should be FiiNcanopy.

54. Fig 10: Show the values with the same number of decimals, even if 0.1 is 0.10.

Response: Yes. We have revised it accordingly.

55. Fig. 11. In my copy it is not easy to differentiate the lines with zenith angle 30 and 0. At least in subpanel c) the legend also looks suspicious. If you want to have a w/b –figure here, could you maybe differentiate the lines with different widths or styles?

Response: Yes. In the revised version, we have used different line styles and linewidths to improve the readability.

56. l. 444: So, are these now leaf level values?

Response: These statements refer to leaf-level results. We have specified this in the revision.

Response: More disperse data points suggest lower correlation between SIF and GPP in the morning. We have discussed one possible reason in section 4.3, which is the fraction of diffuse light. However, we also believe that there are other possible reasons. For example, LUEp is generally more variable in the morning due to the quicker change of environmental conditions, such as air temperature and humanity, whereas LUEf may not change correspondingly. The different response of LUEp and LUEf to the rapidly changing environment may have caused the lower correlation.

Figure 9: Shaded leaves exhibit higher light use efficiency of sustained heat dissipation than sunlit leaves, which is inconsistent with measured results (Figure 7). Briefly discuss the difference.

Response: Thank you for this comment. The model does not simulate a different response to environmental variables between sunlit and shaded leaves (it considers just one type of leaf). The very small difference in simulated heat dissipation is because the sunlit leaves have a higher temperature than the shaded leaves. This affects the sustained heat dissipation in the model. We made a mistake in using the color map and have corrected it.

Line 464 (Figure 10b): The correlation coefficient between PRI and APARcanopy is 0.28, not -0.28 in Figure 10b, right?

Response: The correct value is 0.28. We have revised Fig. 10b in the revision.

In addition, many sentences can be improved, for example: Line 452: per leaf unit area → per unit leaf area

Response: We have checked and improved the overall quality of our writing.

**Short comment from Dr. Kaiyu Guan and our response**

Review for "Unravelling the physical and physiological basis for the solar-induced chlorophyll fluorescence and photosynthesis relationship" by Yang et al.

**General comments**

This manuscript used field measured leaf and canopy fluorescence and photosynthesis and investigated the physical and physiological basis of SIF-GPP relationship at a corn field. They found that APAR dominated the positive SIF-GPP relationship. They further used the continuous active fluorescence measurements from the MoniPAM system to analyze the relationship between fluorescence yield and photochemical yield at leaf scale and found a moderate correlation between the efficiencies of fluorescence emission and photochemistry for sunlit leaves but a weak correlation for shaded leaves. The manuscript has some strength.

The major strengths are: (1) The author combined leaf-scale active fluorescence measurements to fully investigate the physiological basis of the SIF-GPP relationship which is lacking in many studies. (2) The authors are on top of the most recent literatures in this topic. The references used are up to date, and the authors had a very thorough summary of the past literatures. The manuscript is also well-written. However there are several unclear points which should be addressed:

Dear Dr. Guan, Thank you for your positive and encouraging feedback, as well as the clear summary and constructive suggestions. We have revised our manuscript according to your and the other two reviewers' comments.

(1) The reliability of relative efficiency of the sustained heat dissipation ($\Phi D*$) calculation. In L210, the author claims that "Because $Fm$ was measured during the night in the absence of both reversible heat dissipation and photochemistry, a change in $Fm$ must be caused by a change in the sustained heat dissipation". But during night, there are still $\Phi N$ and $\Phi F$ from Fig. 5. I am concerned about the reliability of $\Phi D*$ calculation since to my knowledge, this calculation hasn't been used in previous studies. The author should provide more literature to back up this method.

Response: You are absolutely right that, as far as we know, the derivation of $\Phi D*$ has not been reported in other places. It is also correct that $\Phi F$ is still present in the night because $\Phi F$ is derived from MoniPAM Ft measurements, which are induced by the measuring light. The values are below 100 in the night since leaves are dark-adapted and have maximal $\Phi P$. The nighttime $\Phi N$ is not at the absolute zero, but it is very small (<0.05, and <1% from the pie chart), which is most likely due to the uncertainties in the MoniPAM measurements.

Our idea is that $\Phi N$ is negligible in the night and $\Phi P$ is zero when saturating light is applied. Hence, the change of $\Phi F$ (i.e., Fm) represents the change of $\Phi D$, since $\Phi N + \Phi P + \Phi D + \Phi F = 1$, where $\Phi N \approx 0$ and $\Phi P = 0$. We hope our explanation makes the issue clear.

(2) The data availability across the whole growing season is not provided. In L154, the author mentioned that they excluded 29 days rainy and cloudy data, but the whole period of available canopy data is not provided. The author could provide a time series of the SIF, GPP, APAR data in the supplementary. Also, the availability of the active PAM measurements is also not explicitly provided.

Response: We had provided all the measurements of GPP, SIF, and MoniPAM measurements in a supplement. The link to the data is on the same page with the manuscript below the manuscript pdf icon (https://bg.copernicus.org/preprints/bg-2020-323/bg-2020-323-supplement.zip).

(3) The author reported the overall correlation between $\Phi P canopy$ and $\Phi F canopy$. It would be good that they provide the scatter plot and compare this with the leaf scale relationship.

Response: We have included the suggested plot in the appendix (Fig. A1). We agree that such a plot can give more detailed information to the reader and allow a more informative comparison with the correlation at the leaf-level. Thank you for the suggestion.

(4) L423 They found no clear relationships between $\Phi P canopy$ vs. $\Phi P$ or $\Phi F canopy$ vs. $\Phi F *$. This result needs more explanation, such as this poor correlation is for sunlit leaves or for shaded leaves or both and what causes this poor correlation. Of course, they are from different levels (leaf vs canopy) and canopy structure plays a role here. Although fesc calculation still has large uncertainty, there are several methods proposed to quantify this term (e.g., NIRv/fPAR). The author should try to correct fesc effect and get canopy total $\Phi F canopy$ and compare with leaf $\Phi F*$.

Response: Thank you for this comment. We have included scatter plots of the leaf and canopy efficiencies in the appendix for (both sunlit and shaded) as shown in Fig. A2. Furthermore, we have added a section discussing about the role of fesc on the SIF-GPP relationship. We found that the accuracy of fPAR is crucial to estimate fesc and total emitted SIF when using either FCVI or NIRv. Although we did not find an improvement in GPP estimation after correction TOC SIF for fesc, we believe that canopy total emitted SIF is a better indicator of GPP compared with TOC SIF. With either a better estimation or measurement of fPAR or i0, we can improve the relationship between SIF and GPP by accounting the fesc effects.

(5) L440. They found progressive increase of sustained heat dissipation ($\Phi D *$) during senescence. In contrast with no seasonal variation of $\Phi N$. Why there is no seasonal variation of $\Phi N$? What factor determined the seasonal variation of $\Phi N$.

Response: As shown in the Figs. 4 and 5 (i.e., the seasonal variation of energy partitioning at the leaf level), there was some seasonal variation of $\Phi N$, but its variation has no clear pattern. We think that $\Phi N$ is mainly determined by the radiation levels, which is more pronounced in a diurnal cycle. As in a, seasonal course, the pigment pool of the leaf certainly plays a role since $\Phi N$ or NPQ is related to the carotenoid content. However, as far as we know, the relationship between carotenoid and NPQ is still not clear yet. The challenge is to eliminate the effects of absorbed radiation levels on NPQ. Because the absorbed radiation is also related the pigment pool (e.g., Chl and carotenoid content), it is difficult to separate the effects of pigment content on NPQ from the effects of absorbed radiation. The data we have in this study are not sufficient to give an answer to what factor determined $\Phi N$. To further investigate, we think dedicated laboratory experiments are needed.

(6) L455. The author mentioned that reversible heat dissipation is responsible for the positive relationship between $\Phi F$ and $\Phi P$ at diurnal scale, but there is no diurnal relationship between $\Phi F$ and

$\Phi P$ in the current manuscript. The author only provided the seasonal and seasonal+diurnal relationships.

Response: Thanks for this comment. Indeed, we did not provide the diurnal relationships between $\Phi F$ and $\Phi P$ separately. We have included a figure for their diurnal relationship in the revised manuscript in the appendix (Fig. 8). More positive diurnal correlation between $\Phi F$ and $\Phi P$ are found for sunlit leaves than for shaded leaves. For the correlation $\Phi N$ and $\Phi P$, positive correlations are found for both sunlit and shaded leaves.

(7) L520. The author claimed that a stronger relationship between SIF and GPP for dense canopies is expected since $\Phi F$ sunlit and $\Phi P$ sunlit are moderately correlated. I am not convinced that dense canopy means the fraction of sunlit leaves is larger. Also, the poor correlation between SIF and GPP at senescent stage is probably due to the less data points and more uncertainty of the SIF retrieval.

Response: We agree that the less data points and larger uncertainties of the SIF retrieval are also possible reasons for the lower correlation between SIF and GPP at the senescent stage. We believe that leaves in the upper layer absorb a major part of the incoming PAR, and thus contribute more to TOC SIF and GPP for dense canopies. These leaves are normally sunlit, for which $\Phi F$ and $\Phi P$ are moderately correlated. Dense canopy does not mean that the fraction of sunlit leaves is larger. In fact, the simulations (Fig. 13 in the revised manuscript) show that larger LAI leads to lower sunlit fraction. However, the relevant quantity is fPARsunlit/fPARtot, which supposes to be higher for dense canopies.

(8) L528. The author claimed that under cloudy conditions, SIF-GPP relationship becomes worse. But this is opposite to the previous study from Yang et al. (2018) in a rice paddy. They found similar relationship under sunny and cloudy conditions. Why will diffuse condition lead to a worse SIF-GPP relationship?

Response: Thanks for pointing this out. Indeed, Yang et al. (2018) reported that an identical correlation between SIF and GPP for sunny and cloudy days as indicated by the R2 and rRMSE values (Fig. 4 in Yang et al, 2018). We think that this is not opposite to our results, but suggests that the relationship between SIF and GPP changes under various environmental conditions. The possible cause of a worse SIF-GPP relationship under diffuse (or cloudy) condition, we think, are 1) the higher contribution to TOC SIF from shaded leaves, in which a very weak $\Phi F$-$\Phi P$ relationship occurs, and 2) measurements of TOC SIF are more likely to be more noisy under diffuse illumination in cloudy days.

Yang, K., Ryu, Y., Dechant, B., Berry, J. A., Hwang, Y., Jiang, C., ... & Yang, X. (2018). Sun-induced chlorophyll fluorescence is more strongly related to absorbed light than to photosynthesis at half-hourly resolution in a rice paddy. Remote Sensing of Environment, 216, 658-673.

(9) Overall, I feel that the link between MoniPAM active fluorescence and canopy SIF is weak and the author analyzed these two datasets separately. Although they used to SCOPE but only to model the leaf scale relationship. It would be good if the author can use the leaf measurements to run SCOPE and get canopy SIF and GPP and compare with observations.

Response: We agree with the importance of the link between leaf and canopy measurements. We think that leaf measurements gives the physiological information while canopy measurements are strongly affected by canopy structure. Of course, leaf physiological traits have an impact on the relationship between SIF and GPP on canopy scale. In section 4.3 (i.e., physically and physiologically joint effects on the SIF-GPP relationship), we discuss the link between leaf and canopy measurements by using the two-leaf model. Because we don't have the measurements or estimation of the fraction of sunlit canopy, it is difficult to link the measurements in this study quantitatively. Nevertheless, we discuss the possibility to estimate this fraction once we know some canopy structural parameters.

As for the SCOPE simulation, it is indeed a good way to link leaf and canopy observations. However, to run SCOPE, many more properties of the leaf and canopy structure are required. We have done such an experiment: i) retrieving leaf and canopy structural parameters from canopy reflectance measurements, ii) using the measured leaf physiological traits, and the estimated leaf and canopy structural properties as input to drive the SCOPE model and iii) simulating canopy GPP, SIF and comparing with the measured GPP and SIF. Because many details are required to interpret correctly the experiment and results, we think it is better to present in a separated paper, since

Finally, I want to provide encouragements for this work. The general goal that this work aims to achieve is worth praising. I enjoyed the reading of this manuscript and it clearly shows the authors have been putting lots of efforts into the literature review. I can see that this work could have a good impact and contribution to this field if all the above concerns can be properly addressed. Thus I fully encourage moderate revision of this work. Meanwhile, please understand that a rigorous scrutiny is necessary here as this topic that you are addressing is very important and your conclusion can have a large impact for the general public's understanding about SIF and photosynthesis.

Response: We agree totally with your recommendation and appreciate the constructive comments. We hope that the additional figures and section we added have addressed your concerns. Thank you again!

**Revised manuscript with the changes marked**

[revised manuscript text omitted]

---

## Author Response (AR2)

**Unravelling the physical and physiological basis for the solar-induced chlorophyll fluorescence and photosynthesis relationship using continuous leaf and canopy measurements of a corn crop**

**Response to editor's and reviewers' comments (R2)**

*Peiqi Yang, Christiaan van der Tol, Petya K. E. Campbell, Elizabeth M. Middleton*

*December, 2020*

**R2, Comments from the editor to the Author and our response**

Dear authors,

Thank you for submitting your revised manuscript. As you can see, the reviewers have noted that your work has significantly improved and that you have fully addressed most of the reviewer's comments (with one exception, see Reviewer 1's comment on lines 594-595). I agree that the manuscript has significantly improved. However, Reviewer 1 has a long list of minor comments still. As such, I recommend that your paper be published subject to minor revisions. When you are addressing Reviewer 1's comments, please pay particular attention to the comment about statistical significance. This should be a straightforward addition to make and will make interpretation of the numbers easier for the reader.

Best regards,

Alex Konings

Dear Dr. Konings,

We thank you again for your valuable time. We have addressed the reviewer's comment on l594-595 by extending our answer to Dr. Guan comment #7 in the revised manuscript. Also, we have added the p-values to indicate the significance of the relationship reported in the manuscript. The minor comments have been addressed as well. Please find below our item-to-item response.

Peiqi Yang

**R2, Comments from Anonymous Referee #1 and our response**
**General comments**

I very much enjoyed reading the revised version of the paper. I also think that the authors have done adequate job in answering the comments and the revision.

Response: We thank you again for the constructive comments.

I liked the answer to Reviewer #2 comment (lines 277-279) and would like to see the context also in the manuscript (right now I didn't notice that).

Response: We have included the discussion in the revised manuscript.

"For dense canopies, the leaves in the upper layer absorb a large fraction of incoming radiation, and less radiation can penetrate to lower layer and be absorbed by shaded leaves. This results in that the quantity fAPARsunlit/fAPARtotal is generally higher for dense canopies, such that the contribution of sunlit leaves to the canopy SIF and GPP is higher for dense canopies than for sparse canopies."

One thing that I was thinking during reading, was that very much of this work is based on correlation values, but their significance (p-value) is not addressed at all. The authors could consider adding that.

Response: Thanks for this important comment. We have stated in the revised manuscript that 'the relationship presented in this study was statistically significant (p<0.01) unless otherwise stated.' In the table 2 and 3, we have indicated the insignificant relationship by marking the correlation coefficients italics.

Below some minor comments, that the authors might want to address.

Minor comments (the line numbers refer to the version with the track changes):

l. 32: Need to add reference about the influence of carbon uptake on understanding of climate.

Response: We have added several references to support our statement, namely Falkowski et al., 2000; Friedlingstein, 2015 and Solomon et al., 2009.

l. 35: Now you added 'of net carbon flux', but don't mention that GPP can be further estimated from the net carbon flux.

Response: We have stated that the measurements from eddy covariance can be used to estimate GPP in the revised version.

l. 35: It's not actually 'carbon flux', but mainly the uptake of carbon, not the net flux.

Response: To be consistent with the second sentence in this paragraph, we have changed 'carbon flux' to carbon uptake.

l. 43: Would these two cases be examples, 'e.g' and not the whole research done on this?

Response: Yes, there are examples. We have added 'e.g.,' to indicate that.

l. 45: Why not add references, there are plenty of publications that looked into this?

Response: Thanks for the suggestion. We have added several references to support our statement. 'Damm et al., 2015; Mohammed et al., 2019; Wieneke et al., 2016'.

l. 63: This Guanter study is based on remote sensing whereas the following paragraph deals with in-situ studies. I'd make this distinction now clearer and do you somewhere then also comment, how your finding of this study also possibly contribute to satellite-based observations or if the message from the canopy level is directly applicable also to those?

Response: We agree it is better to point out the study of Guanter is satellite-based, and have specified this by revising the sentence as 'when explaining the SIF-GPP relationship at the satellite level'.

We believe that the message from the canopy level is directly applicable to satellite-based observations. We have added several sentences in the conclusion to address the significance of our study on monitoring photosynthesis from space.

'This study unravels the individual effects of incoming light, vegetation structure and leaf physiology and highlights their joint effects on the relationship between canopy fluorescence and photosynthesis. Our findings on the physical and physiological basis for the SIF and GPP relationship at leaf and canopy levels facilitate the monitoring of photosynthesis from space by using SIF.'

l. 101: Not clear what you mean by 'because of the importance of...'. I don't understand now how that makes study of corn more relevant than other species.

Response: We agree with the reviewer that the study more corn are not more relevant than other species. However, we wanted to address that the investigation of corn is important. In the revised version, we have made our statement clearer by adding 'and encourage more such studies of important crops affecting food security.'

l. 128: NEP: often the instantaneous flux measurement is called NEE and the integrated annual value NEP. You might consider here using NEE instead NEP.

Response: Thanks for the explanation. We have changed NEP to NEE since we talk about measurements over time scales of hours.

l. 131: Where was the FloxBox sensor pointing at and what is its field of view? Was it towards a sunlit area of the vegetation throughout the day?

Response: These TOC measurements were collected from approximately 1.5 m above the canopy at nadir, covering a 25° field of view (0.66 m diameter at ground level). Therefore, the signals observed were a mixture of sunlit and shaded canopy.

The information of the field of view and ground target have been included in the manuscript.

l. 159: Why did you exclude densely clouded days?

Response: We excluded the densely clouded days, because SIF retrieval is generally reliable under clear-sky conditions with only gradual changes in illumination but not under cloudy conditions when large, unpredictable fluctuations of illumination occur. We have added this reason in the revised version.

l. 232: I'm surprised you call here Ci a parameter. Based on the first sentence of the paragraph it would sound like a variable influencing photosynthetic lue…

Response: The reviewer is right. Ci here is a variable but not a parameter. We have changed 'parameters' to 'variables'.

l. 235: Did you do any calculations for the significance of the partial correlation?

Response: Yes, we have done that. The p-values associated with the partial correlation were less than 0.01, except for the two cases in Table 2, which have been indicated in the caption.

l. 295: Now just Fig. 4. Same in line 305.

Response: Thanks. We have changed Figs. 4a and 4b to Fig. 4 accordingly.

l. 295 paragraph/ l. 320: In the first paragraph you say there were only little changes during nighttime and on the next part you mention noticeable changes. Is this contradictory?

Response: We understand the confusion, but the statements are not contradictory. In 295 paragraph, we described the variation during one night, while in 320 paragraph, the day-to-day variation of night values were compared. We have added in 'through the night' in the 295 paragraph to emphasize the time scale. In the 320 paragraph, it has already stated clearly that 'At the seasonal scale'.

l. 305, paragraph: You only talk about the dynamics in response to light levels. Is there any connection with the absolute light levels between the values reached? Or should that even be linked? In some instances the (sunlit) FiiP goes lower with higher light levels, but it's not obvious if this is really taking place… You go more into this in the next section. Maybe you could also say something already here.

Response: There is a general link between absolute light levels and values of PhiP: PhiP decreases with the light levels. However, the exact light response of PhiP (as well as PhiF and PhiN) is controlled by many factors, such as leaf temperature, intercellular $CO_2$ concertation, Vcmax and Jmax. Therefore, instead of linking the light levels and the efficiencies quantitatively, we have provided a summary of the general link and mentioned other controlling factors. The biochemical

model of Van der Tol gives a quantitative link between the environmental factors and the efficiencies of different pathways, but as we discussed, the model itself also has some limitations.

'The light levels largely affected the partitioning of absorbed radiation into the three different pathways. However, other factors, such as leaf temperature, intercellular $CO_2$ concertation and $V_{cmax}$ (which varied seasonally) also played roles in determining the absolute efficiencies of each pathway.'

l. 334: Here the color is denoted as gold, in the caption as yellow.

Response: We have changed the caption to make it consistent with the text.

l. 378: Negatively correlated with each other?

Response: Yes, we have added 'with each other' to be more specific.

l. 391: 'moderately correlated': at light levels above 500 µmol m-2 s-1, or do you mean to dump here together the negative correlation at low light levels and positive correlation at high light levels?

Response: We agree with the reviewer on merging the correlations for the sunlit and shaded leaves in the description. We have revised the sentence as '…was moderately correlated with $\Phi F^*$ at light levels above 500 µmol m-2 s-1 but was negatively correlated $\Phi\_F^*$ at lower light levels '.

l. 418: 'the simulations'?

Response: We have changed 'the simulation' to 'the simulations'

l. 495: What is 'moderately strong'? You could add here the number.

Response: We have added the number ($\rho = 0.53$) as suggested.

l. 592: 'Much larger' than what? Than the shaded?

Response: Yes, we have specified that they were larger than the efficiencies of shaded leaves.

'Therefore, compared with the efficiencies of shaded leaves, $\Phi\_Fsunlit$ and $\Phi\_Psunlit$ have much larger…'

l. 594-595: Sorry, not sure I get it. Dr. Guan also raised this point (#7) and in the reply you discuss this, but it hasn't made it to the manuscript. This point could be clarified.

Response: Indeed, the discussion was not included in the manuscript, but in the new version, we have added the following discussion on the different correlation for dense and spare canopies as following:

For dense canopies, the leaves in the upper layer absorb a large fraction of incoming radiation, and less radiation can penetrate to lower layer and be absorbed by shaded leaves. This results in that the quantity fAPARsunlit/fAPARtotal is generally higher for dense canopies, such that the contribution of sunlit leaves to the canopy SIF and GPP is higher for dense canopies than for sparse canopies.

l. 603-605: But you removed the cloudy days, so can you really see this in your data?

Response: Yes, we removed the densely cloudy and rainy days with rapid change (<~ 2 minutes) in illumination (e.g., rain and clouds on windy days, very low solar angles) because the SIF algorithms do not converge and the calculated values suffer from artefacts.

The discussion here is a general statement about the role of diffuse light by looking at the equation 10 and radiative transfer of diffuse and direct light. In turns of our experiment, we meant that during the day, there was time with cloudy or haze conditions, such as in the early morning.

In the revised version, we have addressed that 'we excluded the data collected on rainy or densely clouded days in the analysis to ensure the quality of SIF retrieval. Nevertheless, the relative fraction of diffuse light is also a possible…'.

l. 623: Ps or Psun?

Response: We used two different terms to differential the sunlit probability at a given depth (x) and the sunlit fraction of the whole canopy. In the revised version, we have decided to use one symbol but clearly wrote 'Ps(x)' to indicate that the quantity is a function of canopy depth x.

l. 676: Sorry, how did you come to equation 13?

Response: We have added some more explanation of Eq. 13: 'it is possible to estimate fesc and canopy total emitted SIF irradiance at 760 nm $F_{760}(tot)$ (i.e., $F_{760}(tot)=\pi \, iPAR \cdot fAPAR \cdot \Phi F_{canopy}$) by correcting radiance of the TOC SIF in the viewing direction ($F_{760}$) for the escape probability'.

l. 689: 'better correlated with Ftot than F'?

Response: Yes, sorry for the typo. We have changed 'and' to 'than'.

Table 2: In the caption you could mention that it is in the first line that you have the correlation coefficient.

Response: We have revised the caption as suggested. 'Correlation coefficients **(the first row)** and partial correlation coefficients (i.e. controlling for or eliminating separate effects) between fluorescence and photosynthesis'

Table 3: You haven't mentioned in the caption the meaning of the value in bold. Are these just high correlations? You could make bold the values that have a significant correlation (small p value).

Response: Thanks for the suggestions. We meant to mark the correlation coefficients greater than 0.70, but in the revised version, we have made italics the values that have a large p value (insignificant correlations).

Fig 4, caption: in my copy it seems to be blue solid lines instead of dashed (as mentioned in the caption).

Response: Yes, it is blue solid lines. We have corrected the caption.

Fig. 8. The first value is on top of the y-axis. It would be recommended to extend the scale of x-axis, so that this doesn't happen.

Response: Agreed. We have revised the figure by extending the x-limit and y-limits.

[revised manuscript text omitted]